# An accurate, precise, and affordable light emitting diode spectrophotometer for drinking water and other testing with limited resources

Michael W. Prairie[1], Seth H. Frisbie[2]*, K. Kesava Rao[3], Anyamanee H. Saksri[2], Shreyas Parbat[3], Erika J. Mitchell[4]

**1** Department of Electrical and Computer Engineering, Norwich University, Northfield, VT, United States of America, **2** Department of Chemistry and Biochemistry, Norwich University, Northfield, VT, United States of America, **3** Department of Chemical Engineering, Indian Institute of Science, Bangalore, Karnataka, India, **4** Better Life Laboratories, Incorporated, East Calais, VT, United States of America

* sfrisbie@norwich.edu

**Data Availability Statement:** All relevant data are within the manuscript and its Supporting Information files.

## Abstract

Spectrophotometers are commonly used to measure the concentrations of a wide variety of analytes in drinking water and other matrixes; however, many laboratories with limited resources cannot afford to buy these very useful instruments. To meet this need, an accurate, precise, and affordable light emitting diode (LED) spectrophotometer was designed and built using best engineering practices and modern circuit design. The cost and performance of this LED spectrophotometer was compared against 4 common commercial spectrophotometers. More specifically, the performance of these spectrophotometers was evaluated from the upper limits of linear range, upper limits of operational range, calibration sensitivities, $R^2$ values, precisions of standards, estimated limits of detection, and percent calibration check standard recoveries for the determinations of iron (Fe), manganese (Mn), and fluoride (F⁻) in drinking water. This evaluation was done in the United States (U.S.) and India. Our LED spectrophotometer costs $63 United States Dollars (USD) for parts. The 4 commercial spectrophotometers ranged in cost from $2,424 to $7,644 USD. There are no practical differences in the upper limits of linear range, upper limits of operational range, $R^2$ values, precisions of standards, and estimated limits of detection for our LED spectrophotometer and the 4 commercial spectrophotometers. For 2 of the 3 analytes, there is a practical difference in the calibration sensitivities our LED spectrophotometer and the 4 commercial spectrophotometers. More specifically, the calibration sensitivities for Mn and F⁻ using our LED spectrophotometer were 65.2% and 67.0% of those using the 4 commercial spectrophotometers, respectively. In conclusion, this paper describes the design, use, and performance of an accurate, precise, and extremely affordable LED spectrophotometer for drinking water and other testing with limited resources.

**Funding:** Non-specific support was provided by Norwich University and the Ministry of Human Resource Development, Government of India. EJM is affiliated with Better Life Laboratories and did not receive support in the form of a salary. Better Life Laboratories also provided no funding for this project. The funders had no role in study design, data collection and analysis, decision to publish, or preparation of the manuscript. The specific roles of these authors are articulated in the 'author contributions' section.

**Competing interests:** EJM's affiliation is with Better Life Laboratories, a nonprofit organization that conducts scientific research and provides technical expertise, equipment, and training to help needy people around the world. Better Life Laboratories received no specific funding for this project from any donors. Donors to Better Life Laboratories provided no input in choosing the subject matter of this project, the materials selected to build the device, the brands or models of equipment selected for comparison, the method of analysis, the research findings, or the manner of disseminating the results. This does not alter our adherence to PLOS ONE policies on sharing data and materials.

## Introduction

Spectrophotometers are highly versatile. They are used to measure the concentrations of a wide range of inorganic, organic, and biological chemicals. For example, they are used to measure the concentrations of harmful metals in drinking water [1, 2], active ingredients in pharmaceutical products [3], and clinically important molecules in humans [4]. Commercial spectrophotometers typically cost over $2,000 United States Dollars (USD), a sum which can make these instruments unattainable for organizations with limited resources. In this project, we describe the development of a spectrophotometer which can be built by laboratory technicians or others competent in basic electronics for $63 USD in parts. This spectrophotometer has performance characteristics comparable to 4 commonly used commercial spectrophotometers.

We have observed a definite need for accurate, precise, and affordable spectrophotometers for drinking water and other testing in countries with limited resources. For example, in 1997 an author of this paper helped make the first national-scale map of arsenic (As) affected drinking water in Bangladesh with a 10-to-15-year-old Hach DR/3-analog spectrophotometer [1, 2]. This map suggested that 45 percent (%) of Bangladesh's area has drinking well water with As concentrations greater than the 0.050-milligrams per liter (mg/L) national standard [1, 2]. This important work was done at the International Centre for Diarrhoeal Disease Research, Bangladesh (icddr,b), an international health research organization and the national cholera hospital in Dhaka, Bangladesh. At the time, the analog spectrophotometer was the most expensive instrument in icddr,b laboratory. In contrast, other hospitals and clinics in Dhaka, the nation's capital, did not have any spectrophotometers at all. Also, there were only 6 drinking water testing laboratories in this resource limited country of approximately 120,000,000 people where at least half the water sources have unsafe concentrations of chemicals [1, 2, 5]. We believe that an accurate, precise, and affordable spectrophotometer for drinking water and other testing in regions with limited resources would greatly benefit public health by making it possible to test drinking water sources for metals and other chemicals by using standard spectrophotometric methods.

Commercial spectrophotometers typically use a tungsten filament lamp as a source of visible and near-infrared radiation. This lamp produces a continuum spectrum from about 350 nanometers (nm) to 2,500 nm. A relatively complicated optical system using lenses, slits, and a diffraction grating is often used to filter this polychromatic radiation to nearly monochromatic radiation. This nearly monochromatic radiation is passed through a glass cuvette with a liquid sample or reference solution to a solid-state detector. The amount of radiation absorbed by the liquid sample or reference solution is used to measure the concentration of a specific analyte [6].

This use of nearly monochromatic radiation to measure the concentration of a specific analyte is needed to limit interferences, that is, false positives from other chemicals with similar absorption spectra that might be in the sample matrix. The optical systems of commercial spectrophotometers are relatively expensive and produce nearly monochromatic radiation with spectral bandwidths from about 4 nm to 20 nm [6]. By comparison, light emitting diodes (LEDs) are relatively inexpensive and have similar spectral bandwidths; for example, a large selection of single-color ultraviolet and visible region LEDs are currently available with emission maxima between 250 nm and 680 nm that have spectral bandwidths from 10 nm to 20 nm [7]. Single-color infrared region LEDs are currently available with emission maxima between 750 nm and 1,070 nm that have slightly broader spectral bandwidths from 23 nm to 60 nm [7]. Therefore, the cost of a spectrophotometer could be significantly reduced by replacing the tungsten filament lamp and associated optical system of commercial spectrophotometers with an assortment of carefully selected LEDs; the challenge is then to ensure that the LED spectrophotometer also be highly accurate and precise.

Educators and their students at the high school, college, and university levels often build simple LED spectrophotometers as part of the chemistry, physics, or electronics curriculum [8–16]. These simple LED spectrophotometers are very useful for education; however, they usually have simple power, source, and signal conditioning circuits, which limit their performance. Moreover, they are not rigorously evaluated through comparison with several different commercial instruments, using different analytes and in different regions of calibration curves.

Some commercial LED spectrophotometers are used for specific chemical analyses; for example, Xerox Corporation designed and built a LED spectrophotometer for use in the output paper path of color printers [17]. The designs of these commercial LED spectrophotometers are not freely available. As with other commercial laboratory equipment, the designs are typically patented, requiring contracts and license payments before others can build them with the same design.

Several open source LED spectrophotometers have been developed through robust design processes for potential use in routine laboratories as well as research or teaching [18–24]. These open source spectrophotometers have been reported to have performance capabilities comparable to those of some commercial spectrophotometers. However, these open-source LED spectrophotometers have been designed and tested for just 1 or 2 analytes, and their performance has usually been compared to that of only 1 to 2 commercial spectrophotometers. These projects have generally not examined the upper limits of linear range, upper limits of operational range, calibration sensitivities, and instrument stability in rigorous detail.

In this article, we describe the design, construction, testing, and use of a new accurate, precise, robust, and extremely affordable LED spectrophotometer and make the plans freely available as open source hardware. This LED spectrophotometer was iteratively designed, tested, and improved over a 3-year period by an international team of professors and students of electrical engineering, analytical chemistry, and chemical engineering. This spectrophotometer uses an advanced signal conditioning chain. Another unique feature of this spectrophotometer is that it can be powered by a 6-volt (V) motorcycle battery; this allows the instrument to be used in remote areas. In this paper, we provide a complete list of parts, schematic diagrams of all circuits, and detailed explanations of circuit design and performance testing so that knowledgeable readers can build, test, and use an exact copy of this LED spectrophotometer themselves, making spectrophotometric methods for analyzing drinking water and other key substances such as biological specimens much more affordable for laboratories with limited resources. Readers can build this accurate, precise, robust, and versatile LED spectrophotometer for approximately $63 USD in parts, less than 2.6% of the cost of several commercial spectrophotometers.

Our diverse international team rigorously evaluated this spectrophotometer by comparing its performance with that of 4 different commercial instruments, for 3 different analytes, and in different regions of the calibration curves. Additional novel aspects of this study are the use of higher order polynomial relationships to objectively test the linearity of calibration curves [25] and the use of control charts to estimate detection limits [26]. This new LED spectrophotometer shows performance characteristics comparable to those of 4 commercial spectrophotometers for testing Fe, Mn, and $F^-$ within ranges commonly found in drinking water.

## Materials and methods

The requirements, specifications, and component selection choices are summarized in Fig 1. The details of the operation of the various parts are included in the system description, so only summary decisions are included here. Unless otherwise stated, all passive components were easily available resistors with at most 5% tolerance, and capacitors with at most 20% tolerance.

This LED spectrophotometer has 4 subsystems: (1) a power circuit, (2) a LED source circuit, (3) a sample holder, and (4) a photodiode (PD) detector circuit (Figs 2 and 3). A list of the parts and associated prices for building this LED spectrophotometer are shown in Table 1.

The goal of this project was to build a robust spectrophotometer whose performance would be comparable to commercial spectrophotometers while keeping the cost as low as possible. Model numbers of components, suppliers, and costs are presented here to show how much money we paid for these components, but actual costs may vary according to suppliers. Unfortunately, as pointed out by an anonymous reviewer, some components may cost more in resource-limited countries due to limited availability.

## Power circuit

The LED spectrophotometer requires a 6-volt (V) minimum to 12-V maximum direct current (DC) input to the power circuit. If there is no municipal power, this 6-V minimum allows the

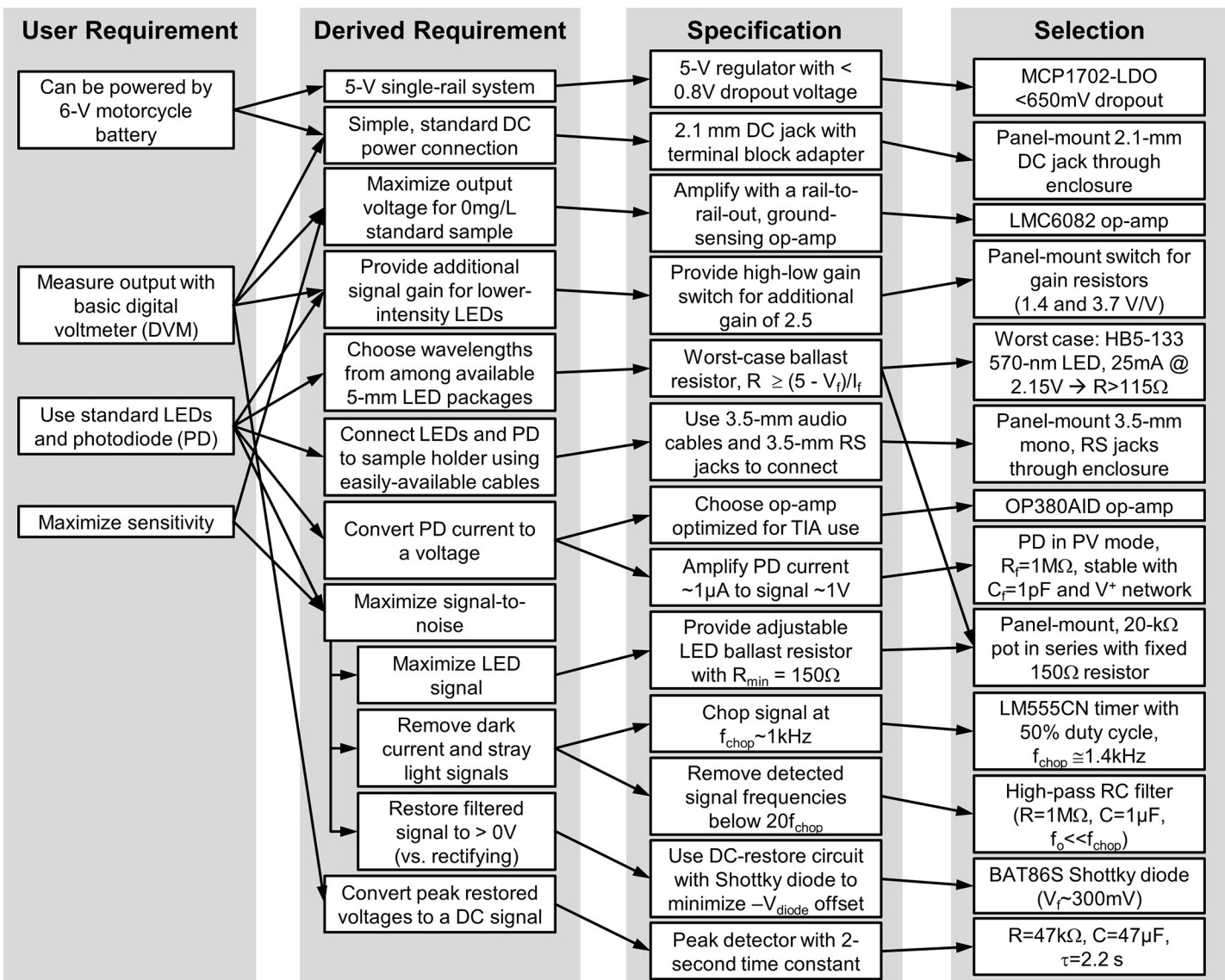

**Fig 1. The requirements, specifications, and component selection of the light emitting diode (LED) spectrophotometer.**

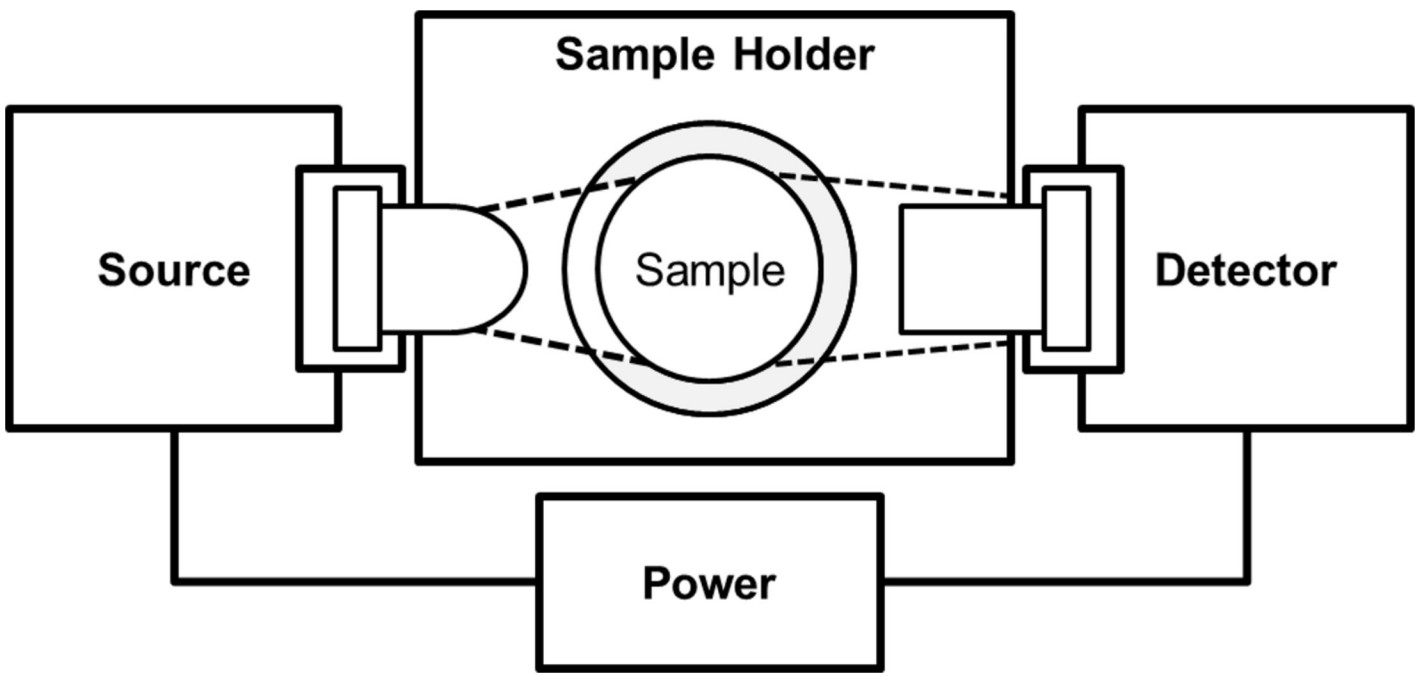

**Fig 2. A block diagram of the light emitting diode (LED) spectrophotometer.**

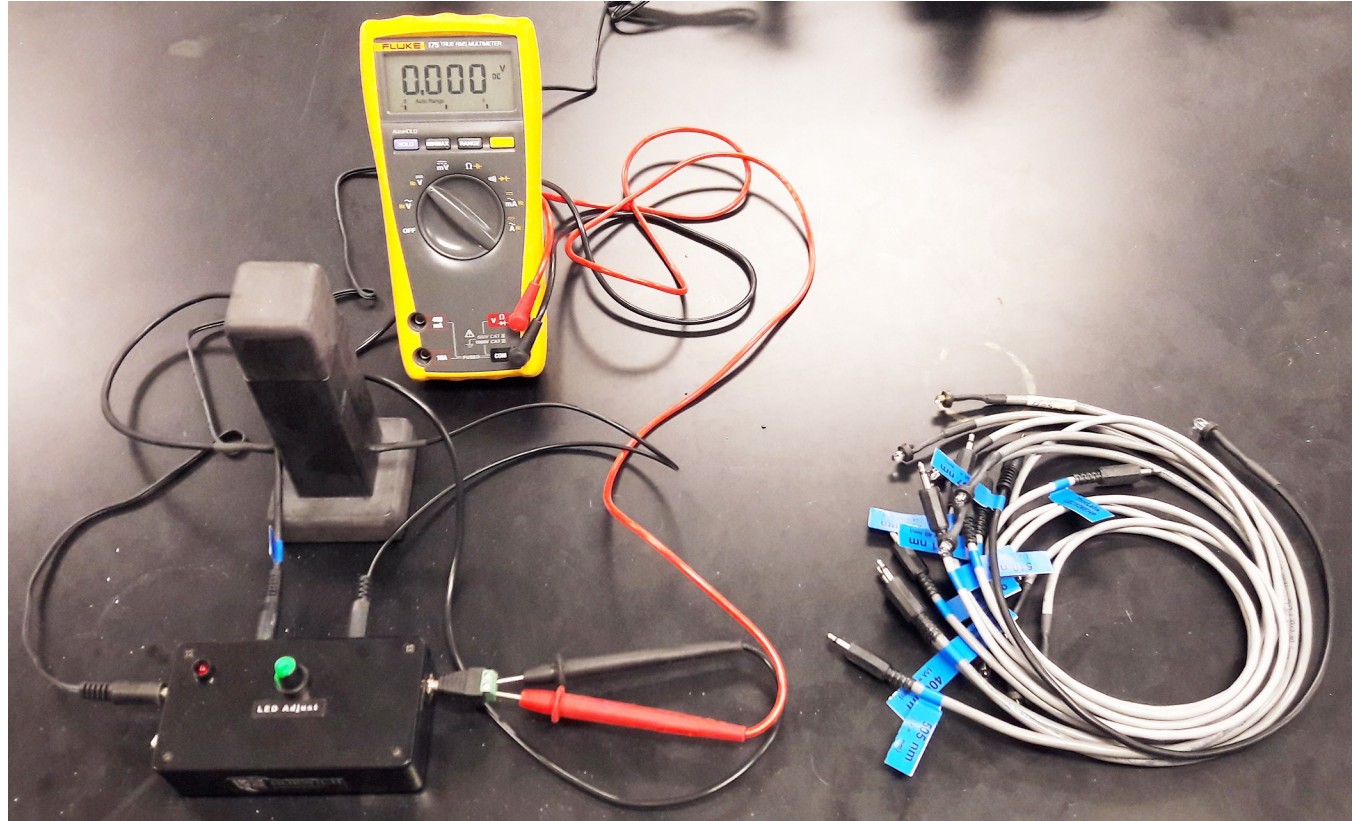

**Fig 3. A photograph of the light emitting diode (LED) spectrophotometer.**

**Table 1. A list of the parts and associated prices in current (2019) United States Dollars (USD) for 1 light emitting diode (LED) spectrophotometer.** These prices are from Newark (www.newark.com), Digi-Key Electronics (www.digikey.com), and Amazon.com, Inc. (www.amazon.com).

| Part | Value | Device | Package | Description | Price |
|---|---|---|---|---|---|
| C1 | 1 picofarad (pF) | 561R10TCCV10 | Radial, 6.4-millimeter (mm) pitch | Capacitor | 0.79 |
| C2 | 1 microfarad (μF) | MCCB1E105M2ACB | Radial, 5.08 mm pitch | Capacitor | 0.58 |
| C3 | 0.1 μF | MCDTR10M35-1-RH | Radial, 2.5 mm pitch | Capacitor | 0.75 |
| C4 | 0.01 μF | 562R5GAS10 | Radial, 6.4 mm pitch | Capacitor | 0.65 |
| C5 | 33 pF | 561R10TCCQ33 | Radial, 6.4 mm pitch | Capacitor | 1.07 |
| C6 | 47 μF | ECA-1HM470 | Radial, 2.5 mm pitch | Polarized capacitor | 0.25 |
| C7 | 10 μF | EEA-FC1E100 | Radial, 1.5 mm pitch | Polarized capacitor | 0.30 |
| C9 | 1 μF | MCCB1E105M2ACB | Radial, 5.08 mm pitch | Capacitor | 0.58 |
| C14 | 0.1 μF | MCDTR10M35-1-RH | Radial, 2.5 mm pitch | Capacitor | 0.75 |
| C15 | 0.1 μF | MCDTR10M35-1-RH | Radial, 2.5 mm pitch | Capacitor | 0.75 |
| C16 | 0.1 μF | MCDTR10M35-1-RH | Radial, 2.5 mm pitch | Capacitor | 0.75 |
| C18 | 0.1 μF | MCDTR10M35-1-RH | Radial, 2.5 mm pitch | Capacitor | 0.75 |
| C19 | 0.1 μF | MCDTR10M35-1-RH | Radial, 2.5 mm pitch | Capacitor | 0.75 |
| D1 | | BAT86 | DO-34 (SOD68) | Schottky diode | 0.39 |
| D2 | | BAT86 | DO-34 (SOD68) | Schottky diode | 0.39 |
| IC1 | | OPA380AID | 8-SOIC | Transimpedance amplifier | 5.46 |
| IC2 | | LMC6082IN/NOPB | 8-DIP | Operational amplifier | 4.69 |
| IC3 | | LM1117T-5.0/NOPB | TO-220 | Voltage regulator | 1.54 |
| IC4 | | LMC555CN/NOPB | 8-DIP | CMOS timer | 1.35 |
| LED | | Green LED—520 nanometer (nm) | T-1 ¾ | 520 nm (one of several wavelengths) | 0.32 |
| LED_ON | | SSL-LX5093LID | T-1 ¾ | LED | 0.45 |
| ON/OFF | | RA11131100 | Panel mount | SPST rocker switch | 0.49 |
| GAIN_SW | | 1MS1T1B5M1QE | Panel mount | SPDT toggle switch | 1.40 |
| PD | | PDB-C139 | Radial, 5 mm diameter (T 1 3/4) | Photodiode | 4.82 |
| Q1 | | 2N7000 | TO-92 | N-Channel MOSFET | 0.43 |
| R1 | 1 megaohm (MΩ) | MCF 0.25W 1M | Axial leaded | Resistor | 0.09 |
| R2 | 1 MΩ | MCF 0.25W 1M | Axial leaded | Resistor | 0.09 |
| R3 | 1 MΩ | MCF 0.25W 1M | Axial leaded | Resistor | 0.09 |
| R4 | 47 kilo-ohm (kΩ) | MCF 0.25W 47K | Axial leaded | Resistor | 0.09 |
| R5 | 5.6 kΩ | MCF 0.25W 5K6 | Axial leaded | Resistor | 0.09 |
| R6 | 2.2 kΩ | MCF 0.25W 2K2 | Axial leaded | Resistor | 0.09 |
| R7 | 5.1 kΩ | MCCFR0W4J0512A50 | Axial leaded | Resistor | 0.01 |
| R8 | 150 Ω | MCCFR0W4J0151A50 | Axial leaded | Resistor | 0.01 |
| R9 | 1 kΩ | MCF 0.25W 1K | Axial leaded | Resistor | 0.09 |
| R10 | 20 kΩ POT | P160KN-0QC15B20K | Panel mount | Potentiometer | 0.79 |
| R11 | 15 kΩ | MCCFR0W4J0153A50 | Axial leaded | Resistor | 0.07 |
| | | Various vendors | 8-DIP | 8-SOIC to 8-DIP adapter | < 1.00 |
| | | MC002825 | Panel mount | 2.1 mm DC power jack | 0.58 |
| | | MC002825 | Panel mount | 2.1 mm DC power jack | 0.58 |
| | | MJ-074N | Panel mount | 3.5 mm Audio jack | 1.54 |
| | | MJ-074N | Panel mount | 3.5 mm Audio jack | 1.54 |
| | | PSG03744 | Cable | 3.5 mm Stereo male-male cable (cut in half) | 0.57 |
| | | CLB-JL-8111 | Accessory | 2.1 mm plug to terminal adapter | 2.00 |
| | | PPW00026 | Accessory / cable | Male DC 2.1 mm pigtail | 1.15 |
| | | 28–965 | Enclosure | Plastic project box | 4.04 |
| | | MRG1791.0050 | Cable | RG-178U coax (12 inches; for PD cable) | .40 |

*(Continued)*

**Table 1.** (Continued)

| Part | Value | Device | Package | Description | Price |
|------|-------|--------|---------|-------------|-------|
| | | 38K6105 | Jack assembly | 3.5 mm Mono male jack (for PD cable) | 0.93 |
| | | MC72601S | Knob for 6 mm shaft | Rotatory potentiometer knob | 0.75 |
| | | A104800AAC | Panel mount | 5 mm LED holder | 1.80 |
| | | 503 | Copper clad FR4 | Single-sided printed circuit board | 4.65 |
| | | 437 | Moldable silicone rubber | Sugru moldable glue | 2.28 |
| | | F-3599 | 9.525 mm Square | Bumper / feet, stick on (8) | 4.66 |
| | | 09300 | Machine screw, 4–40, 6.35 mm | PCB mounting hardware (4) | 0.40 |
| | | SK00-0044-AKS | Screw nuts, 4–40 | PCB mounting hardware (4) | 0.32 |
| | | DM3-FASTWAZ100DIN125 | Washer, 3 mm ID | PCB mounting hardware (8) | 0.07 |
| | | 24–14687 | 22 AWG stranded wire | PCB to panel component (estimated cost) | 0.50 |
| | | HS511-1.22M | 3 mm | Heat shrink tubing (estimated cost) | 0.10 |
| | | | 0.75 x 1.5 x 12 inch 0.75 x 3.5 x 3.5 inch | Wood material for sample holder (estimated cost) | 1.10 |
| | | 75H5807 | Nylon cable clamp, 0.5 inch | Sample holder spring | 0.08 |
| | | | | Total cost | 63.00 |

LED spectrophotometer to run off a 6-V motorcycle battery. The power circuit provides a stable and constant 5-V regulated DC output to the LED source and PD detector circuits regardless of the 6-V to 12-V input voltage (Fig 4). The design requirement to run off a 6-V motorcycle battery called for a low-dropout (LDO) linear regulator to accommodate the 1-V difference between the 6-V minimum input and 5-V regulated output voltages. A standard linear regulator typically requires a 2-V difference between the minimum input and regulated output voltages [27, 28]. A LM1117-LDO regulator was selected to match these design requirements for the power circuit (Table 1; Fig 4). This regulator has an input voltage that ranges from a 5.35-V minimum to a 13.2-V maximum, a dropout voltage of 650 millivolt (mV), and a maximum current rating of 250 milliamperes (mA).

The power circuit has a manual on-off switch (Table 1; Fig 4). If the power circuit is on, an indicator LED is automatically illuminated (Table 1; Fig 4). The capacitors reduce noise by shunting high-frequency noise from the positive terminal to the ground terminal (Table 1; Fig 4). The 1-kilo-ohm (kΩ) ballast resistor limits the current to the indicator LED (Table 1; Fig 4). The forward voltage of the indicator LED is approximately 2 V; therefore, the 1-kΩ ballast resistor limits the current to approximately 3 mA (Fig 4).

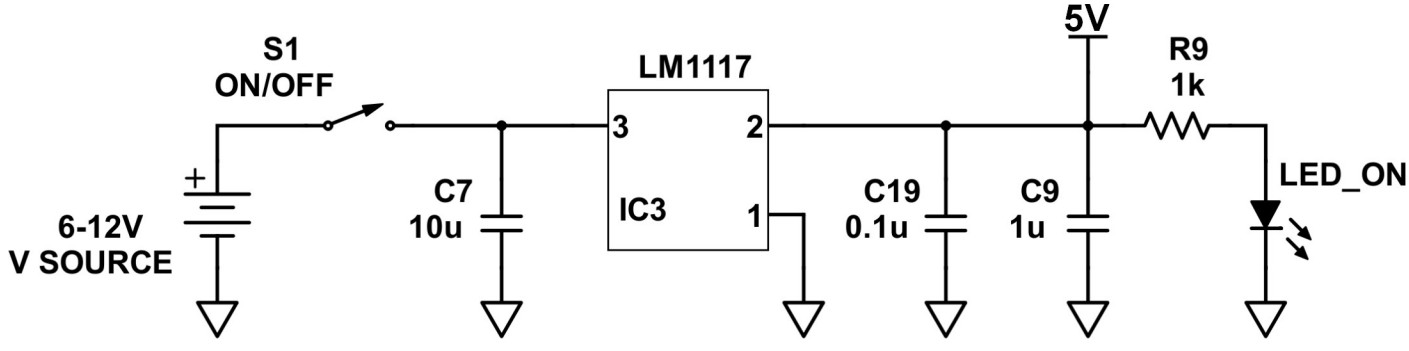

**Fig 4. A schematic diagram of the power circuit.**

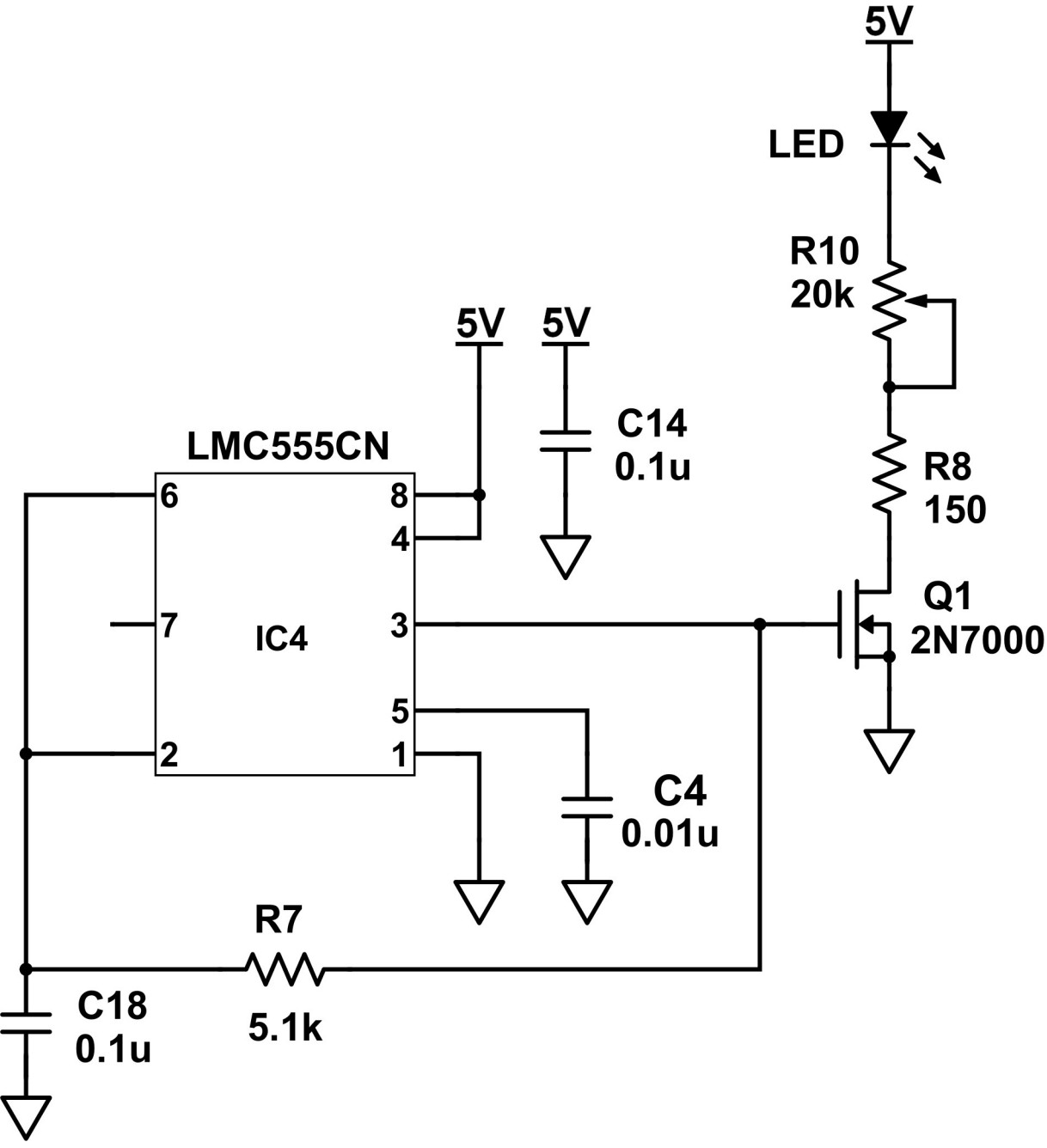

**Fig 5. A schematic diagram of the light emitting diode (LED) source circuit.**

### Light emitting diode source circuit

The source LED is powered by the 5-V regulated DC output of the power circuit. The source LED is controlled by a driver circuit that uses a LM555CN timer circuit (integrated circuit) and a 2N7000-metal-oxide-semiconductor field-effect transistor (MOSFET; Table 1; Fig 5) [28].

The timer circuit is used to generate a square wave of voltage to the transistor (Fig 5). More specifically, the LM555CN timer circuit provides a 50% duty cycle (the voltage to the transistor

is 50% on and then 50% off during each cycle) at a 1.4-kilohertz (kHz) frequency [28]. This duty cycle and chopping frequency are determined by the 5.1-kΩ resistor (R7) and 0.1-micro-farad (μF) capacitor (C18; Table 1; Fig 5) [28]. Capacitor C4 is a 0.01-μF bypass capacitor at the control voltage pin (5) of the timer circuit (Table 1; Fig 5). Capacitor C14 is a 0.1-μF bypass capacitor at the connection to the power circuit (Table 1; Fig 5).

The 2N7000-MOSFET transistor is used as a switch to turn the source LED on and off in response to the square wave of voltage from the timer circuit (Fig 5). Therefore, the source LED generates an on-off square wave of emitted light through the sample holder and to the PD detector circuit (Figs 2, 3 and 5). If the source LED is on, the detector signal results from the light that is transmitted through the sample holder, any stray light from the environment, and any dark current from the random generation or recombination of electron-hole pairs in the photo-diode [29]. If the source LED is off, the detector signal results from any stray light, and any dark current [29]. Therefore, stray light and dark current produce a constant DC voltage shift during both the on and off portions of the detector signal. The source LED is modulated at 1.4 kHz so that this DC voltage shift can be removed by a high-pass filter in the PD detector circuit.

A fixed 150-ohm (Ω) ballast resistor (R8) limits the maximum forward current to 30 mA and prevents the source LED from burning out (Table 1; Fig 5). This fixed ballast resistor (R8) is connected in series with a potentiometer (R10; Table 1; Fig 5). The potentiometer (R10) has a 20-kΩ maximum resistance and is used to adjust the brightness of the source LED and pre-vent saturating the PD detector circuit (Table 1; Fig 5). A typical forward voltage is about 3.5 V when the source LED is on.

The source LED was selected so that the wavelength of the emission maximum of the source LED corresponded to the wavelength of the absorption maximum of the colorized analyte. For example, a source LED with an emission maximum at 510 nanometers (nm) was used for the analysis of a colorized analyte with an absorption maximum at 510 nm. The spectral width of this source LED is about 25 nm, according to the manufacturer (Table 1).

## Sample holder

The sample holder is shown in Figs 2, 3 and 6. It is made mostly of wood (Fig 3). It holds a standard 1.0-centimeter (cm) diameter silicate glass sample cell (Fig 6). It uses a modified nylon cable clamp as a spring to reproducibly hold the sample cell in the sample holder (Fig 6). A wooden cap prevents stray ambient light from entering the sample holder (Fig 6). Finally, the interior of the sample holder is painted black to absorb stray light (Fig 3).

## Photodiode detector circuit

The detector PD is powered by the 5-V regulated DC output of the power circuit. The PD is connected to a signal conditioning chain. This chain uses a transimpedance amplifier, a high-pass filter, a DC-restore circuit, a peak detector, and a final stage amplifier (Fig 7).

The PDB-C139-photodiode is used in photoconductive mode to convert the light that is transmitted through the sample cell to a signal current (PDB-C139; Table 1). The power of this incident light at the PD is directly proportional to the signal current that is output by the PD. This incident power at the PD is roughly 5 microwatts (μW) or less. The responsivity of the PD depends on the wavelength of absorbed light and ranges from approximately 0.15 to 0.25 ampere/Watt (A/W). Therefore, the signal current is roughly 1 microamperes (μA) or less as shown in Eq 1.

$$\lesssim 5 \text{ μW} \times \frac{1 \text{ W}}{10^6 \text{ μW}} \times \frac{\sim 0.2 \text{ A}}{1 \text{ W}} \times \frac{10^6 \text{ μA}}{1 \text{ A}} \lesssim 1 \text{ μA} \tag{1}$$

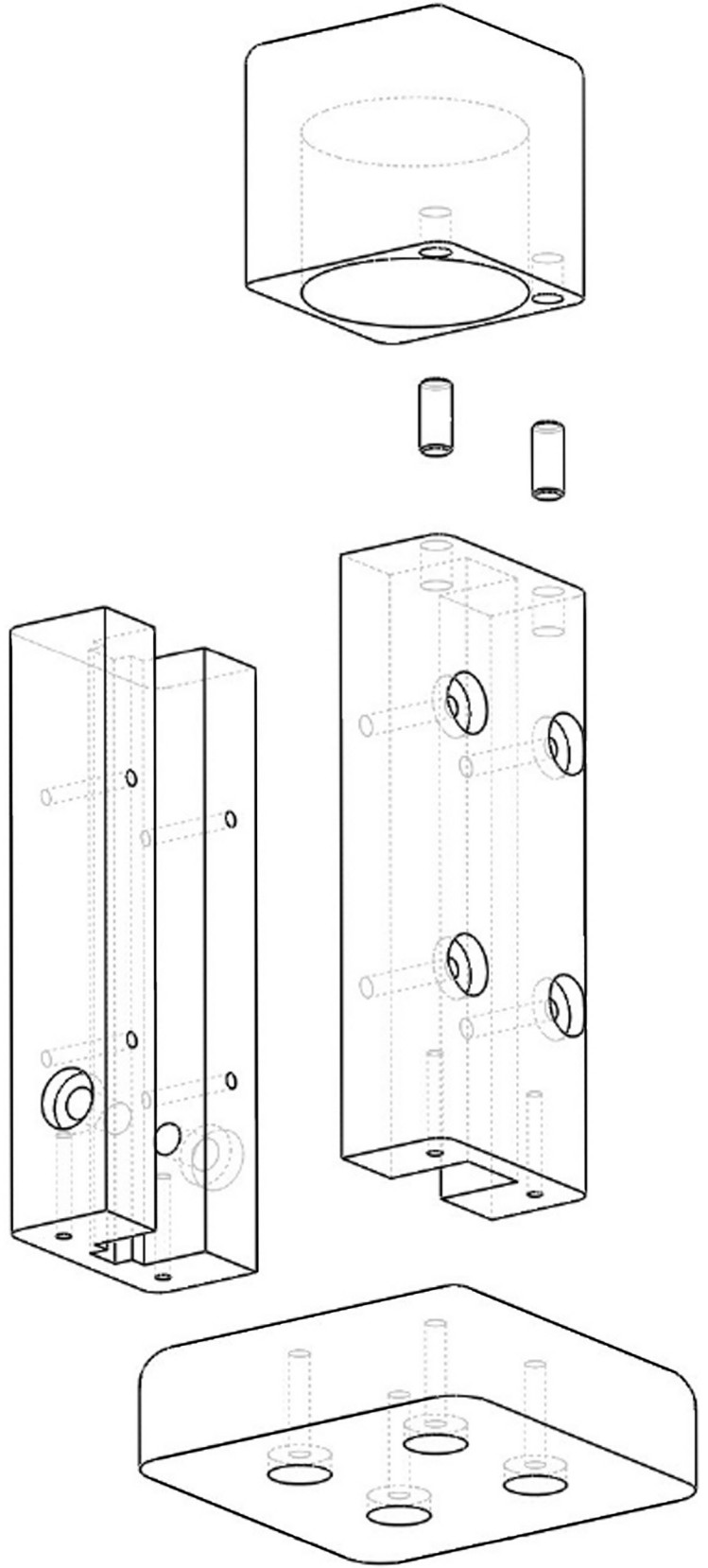

**Fig 6. An exploded view of the sample cell holder.**

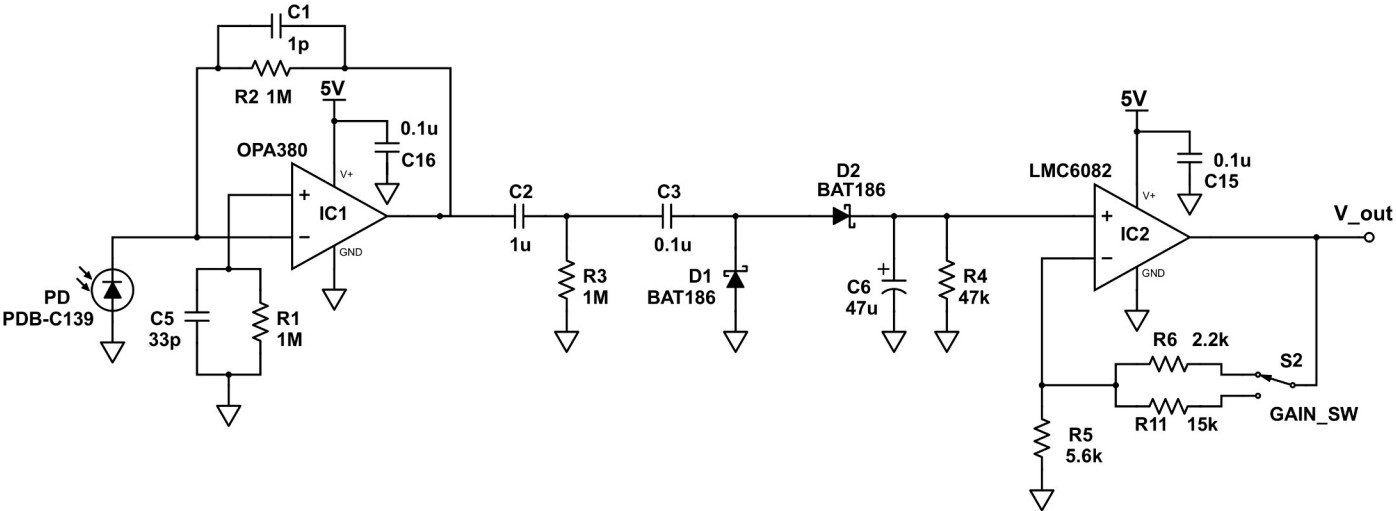

**Fig 7. A schematic diagram of the photodiode (PD) detector circuit.** The signal conditioning chain uses a transimpedance amplifier, a high-pass filter, a DC-restore circuit, a peak detector, and a final stage amplifier.

The OPA380AID operational amplifier is optimized for use as a transimpedance amplifier (TIA; Table 1; Fig 7). The TIA converts the relatively small signal current from the PD to a usable signal voltage. The TIA has relatively small input bias currents, relatively small input offset voltages, and a relatively large gain bandwidth. The gain of the TIA is set by a 1-mega-ohm (MΩ) feedback resistor (R2; Table 1; Fig 7). More specifically, a transimpedance gain of $v_0/i_i = 10^6$ V/A from this 1-MΩ feedback resistor (R2) provides a usable signal voltage that is on the order of 1 V from the relatively small signal current that is on the order of 1 μA as shown in Eq 2.

$$1 \ \mu A \times \frac{1 \ A}{10^6 \ \mu A} \times \frac{10^6 \ V}{1 \ A} = 1 \ V \tag{2}$$

The 1-picofarad (pF) capacitor (C1) that is in parallel with the 1-MΩ feedback resistor (R2) is used to reduce noise (Table 1; Fig 7). More specifically, this capacitor (C1) compensates for the capacitance of the PD introducing a pole in the feedback path at high frequencies; this prevents oscillation of the signal (Fig 7). Similarly, the 33-pF capacitor (C5) in parallel with the 1-MΩ resistor (R1) is also used to reduce noise (Table 1; Fig 7). More specifically, this capacitor (C5) and resistor (R1) compensate for the small input bias currents and offset voltages by matching the impedance at the positive and negative input terminals of the operational amplifier (Fig 7).

The TIA output signal voltage has a small, positive offset from stray light, and dark current. This offset is removed with a high-pass filter, and a DC-restore circuit. The high-pass filter shifts the average value of the square wave detector signal to 0 V; it uses a 1-microfarad (μF) capacitor (C2) and a 1-MΩ resistor (R3; Table 1; Fig 7). The cutoff frequency ($f_{HPF}$) of this high-pass filter is 0.16 Hz; that is, this filter passes signals greater than 0.16 Hz, and attenuates signals less than 0.16 Hz (Eq 3).

$$f_{HPF} = \frac{1}{2\pi C_2 R_3} = \frac{1}{\left(2 \times \pi \times 1 \ \mu F \times \frac{1 \ F}{10^6 \ \mu F} \times \frac{\frac{1 \ s}{\Omega}}{1 \ F} \times 1 \ M\Omega \times \frac{10^6 \ \Omega}{1 \ M\Omega}\right)} = 0.16 \ Hz \tag{3}$$

This DC-restore circuit is a diode clamping circuit; it shifts the minimum value of the square

wave detector signal to 0 V [30]. Therefore, the output of the DC-restore circuit is a square wave of signal voltages that ranges from approximately 0 V to some positive voltage, depending on the concentration of analyte in the sample cell [30]. This DC-restore circuit uses a 0.1-μF capacitor (C3) and a BAT86-Schottky diode (D1; Table 1; Fig 7). When the input signal voltage is negative, the Schottky diode (D1) is forward-biased, the 0.1-μF capacitor (C3) charges, and the output signal voltage is 0 V minus the small forward voltage drop of the Schottky diode (Fig 7) [30]. When the input signal voltage is positive, the Schottky diode (D1) is reverse-biased, the 0.1-μF capacitor (C3) retains its charge, and the output signal voltage equals the input signal voltage plus the capacitor voltage (D1; Fig 7) [30]. Therefore, the dark current and stray light signal are removed, and the full magnitude of the signal voltage is restored [31].

The peak detector circuit uses the Schottky diode (D2), a 47-μF capacitor (C6), and a 47-kΩ resistor (R4; Table 1; Fig 7). The capacitor (C6) is charged by the peak signal voltage (Fig 7). The resistor (R4) allows the capacitor (C6) to discharge with a time constant of 2.2 second (s) as shown in Eq 4 (Fig 7).

$$47\ \mu F \times \frac{1\ F}{10^6\ \mu F} \times \frac{\frac{1\ s}{\Omega}}{1\ F} \times 47\ k\Omega \times \frac{10^3\ \Omega}{1\ k\Omega} = 2.2\ s \tag{4}$$

The final stage amplifier uses a LMC6082-non-inverting operational amplifier with a switchable gain of 1.4 or 3.7 V/V (Table 1; Fig 7). This is a rail-to-rail input, ground-sensing operational amplifier in a single-rail system. The upper rail is the 5-V system voltage, and the lower rail is the 0-V ground reference voltage. This is critical since the peak signal voltage from highly concentrated samples will be close to 0 V; this rail-to-rail input, ground-sensing capability helps minimize error from measuring the small differences in these voltages. The lower 1.4 V/V gain option is used if the maximum intensity of the source LED is sufficient to maximize the TIA output. The higher 3.7-V/V gain option is used if the maximum intensity of the source LED is not sufficient to produce a large TIA output. The final signal voltage is read with a standard voltmeter (Figs 3 and 7).

## Calculating price

A list of the parts and associated prices for building one of the 2 identical LED spectrophotometers used in this study are shown in Table 1. The 3 commercial spectrophotometers used for this study in the United States (U.S.), a Hach DR-2700, a Hach 2010, and a Thermal Electron Spectronic 20D+ are no longer manufactured [32, 33]. So that comparisons of prices in current (2019) USD could be made, the historical manufacturer's suggested retail price (MSRP) for each of these instruments was adjusted for inflation with the consumer price index (CPI) inflation calculator [34]. These prices are shown in Tables 2 and 3. In contrast, the 1 commercial ultraviolet-visible (UV-Vis) absorption spectrophotometer used for this study in India, an Agilent Technologies Cary 60-UV-Vis is currently manufactured [35]. The price for this instrument in current (2019) USD is shown in Table 4.

## Total iron (Fe) by the 1,10-phenanthroline method

The concentrations of total iron (Fe) were determined by the 1,10-phenanthroline method [36]. The absorbance of the complexed Fe product was measured at 510 nm with a Hach DR-2700, a Hach 2010, a Thermal Electron Spectronic 20D+, and 2 identical LED spectrophotometers (Table 2; Figs 8–10). Standard solutions at 0.00, 1.01, 2.00, 3.00, 3.99, 4.99, 5.99, 6.99, 7.98, 8.98, and 9.98 mg/L of total Fe were used to calculate linear range [25], operational range [36], and calibration sensitivity [6, 38]. Seven 0.00-mg/L standard solutions were analyzed as samples and used to calculate the precision of standards [36], and the estimated limit of

**Table 2. The price and performance of a Hach DR-2700, a Hach 2010, a Thermal Electron Spectronic 20D+, and 2 identical light emitting diode (LED) spectrophotometers for the determination of total iron (Fe) by the 1,10-phenanthroline method [36].** $V_0$ = blank voltage.

| Price and Performance | Hach DR-2700 | Hach 2010 | Spectronic 20D+ | LED Spectrophotometer #1 | | | | LED Spectrophotometer #2 | | | |
|---|---|---|---|---|---|---|---|---|---|---|---|
| | | | | $V_0 = 4.364$ | $V_0 = 3.872$ | $V_0 = 3.364$ | $V_0 = 2.842$ | $V_0 = 4.364$ | $V_0 = 3.872$ | $V_0 = 3.366$ | $V_0 = 2.846$ |
| Price in current (2019) United States Dollars (USD) | 3,442 | 2,907 | 2,424 | 63 | 63 | 63 | 63 | 63 | 63 | 63 | 63 |
| Upper limit of linear range (mg/L) | 3.00 | 6.99 | 3.00 | 3.00 | 3.99 | 3.00 | 3.00 | 4.99 | 3.99 | 3.00 | 3.99 |
| Upper limit of operational range (mg/L) | 7.98 | 7.98 | 7.98 | 6.99 | 6.99 | 6.99 | 6.99 | 7.98 | 7.98 | 7.98 | 7.98 |
| Calibration sensitivity (L/mg) using linear regression through the origin for the 0.00, 1.01, 2.00, and 3.00 mg/L standards | 0.174 | 0.180 | 0.165 | 0.138 | 0.136 | 0.138 | 0.138 | 0.163 | 0.165 | 0.167 | 0.169 |
| $R^2$ for the linear regression through the origin for the 0.00, 1.01, 2.00, and 3.00 mg/L standards | 0.99999 | 1.00000 | 0.99994 | 0.99992 | 0.99998 | 0.99992 | 0.99980 | 0.99999 | 0.99999 | 0.99999 | 0.99997 |
| Precision of standards (mg/L) | 0.002 | 0.003 | 0.002 | 0.005 | 0.004 | 0.003 | 0.005 | 0.007 | 0.005 | 0.009 | 0.009 |
| Estimated limit of detection (mg/L) | 0.003 | 0.004 | 0.003 | 0.006 | 0.005 | 0.004 | 0.006 | 0.008 | 0.006 | 0.010 | 0.010 |
| % Calibration check standard recovery | 99.6 | 99.8 | 99.6 | 99.4 | 99.1 | 99.2 | 99.0 | 100.0 | 99.3 | 99.2 | 98.8 |

detection [26]. Lastly, a 3.00-mg/L standard solution was analyzed as a sample and used to calculate the % calibration check standard recovery [36]. This work was done at Norwich University in Northfield, Vermont (VT).

## Total manganese (Mn) by the 1-(2-pyridylazo)-2-naphthol method

The concentrations of total manganese (Mn) were determined by the 1-(2-pyridylazo)-2-naphthol (PAN) method [37]. The absorbance of the complexed Mn product was measured at 560 nm with a Hach DR-2700, a Hach 2010, a Thermal Electron Spectronic 20D+, and a LED spectrophotometer (Table 3; Figs 11 and 12). Standard solutions at 0.000, 0.100, 0.200, 0.300, 0.400, 0.500, 0.600, 0.700, 0.800, 0.900, and 1.000-mg/L of total Mn were used to calculate linear range [25], operational range [36], and calibration sensitivity [6, 38]. Seven 0.000-mg/L standard solutions were analyzed as samples and used to calculate the precision of standards [36], and the estimated limit of detection [26]. Lastly, a 0.300-mg/L standard solution was analyzed as a sample and used to calculate the % calibration check standard recovery [36]. This work was done at Norwich University in Northfield, VT.

**Table 3. The price and performance of a Hach DR-2700, a Hach 2010, a Thermal Electron Spectronic 20D+, and a light emitting diode (LED) spectrophotometer for the determination of total manganese (Mn) by the Hach 1-(2-pyridylazo)-2-naphthol (PAN) method [37].** $V_0$ = blank voltage.

| Price and Performance | Hach DR-2700 | Hach 2010 | Spectronic 20D+ | LED Spectrophotometer #1 | | | |
|---|---|---|---|---|---|---|---|
| | | | | $V_0 = 1.989$ | $V_0 = 1.553$ | $V_0 = 1.011$ | $V_0 = 0.529$ |
| Price in current (2019) United States Dollars (USD) | 3,442 | 2,907 | 2,424 | 63 | 63 | 63 | 63 |
| Upper limit of linear range (mg/L) | 0.500 | 0.600 | 0.700 | 0.500 | 0.600 | 0.500 | 0.400 |
| Upper limit of operational range (mg/L) | $\geq 1.000$ | $\geq 1.000$ | $\geq 1.000$ | $\geq 1.000$ | $\geq 1.000$ | $\geq 1.000$ | $\geq 1.000$ |
| Calibration sensitivity (L/mg) using linear regression through the origin for the 0.000, 0.100, 0.200, and 0.300 mg/L standards | 0.798 | 0.828 | 0.781 | 0.558 | 0.542 | 0.532 | 0.463 |
| $R^2$ for the linear regression through the origin for the 0.000, 0.100, 0.200, and 0.300 mg/L standards | 0.99995 | 0.99997 | 0.99987 | 0.99961 | 0.99943 | 0.99948 | 0.99990 |
| Precision of standards (mg/L) | 0.0006 | 0.0006 | 0.003 | 0.001 | 0.005 | 0.002 | 0.003 |
| Estimated limit of detection (mg/L) | 0.0007 | 0.0007 | 0.003 | 0.001 | 0.006 | 0.002 | 0.004 |
| % Calibration check standard recovery | 98.8 | 99.2 | 107.7 | 98.3 | 99.0 | 99.2 | 98.4 |

**Table 4. The price and performance of an Agilent Technologies Cary 60 UV-Vis (ultraviolet-visible) absorption spectrophotometer, and a light emitting diode (LED) spectrophotometer for the determination of total fluoride (F⁻) by the sodium 2-(parasulfophenylazo)-1,8-dihydroxy-3,6-naphthalene disulfonate (SPADNS) method [36].** $V_0$ = blank voltage.

| Price and Performance | Agilent Technologies Cary 60 UV-Vis | LED Spectrophotometer #1 | | | | | |
|---|---|---|---|---|---|---|---|
| | | $V_0 =$ 0.627 | $V_0 =$ 0.550 | $V_0 =$ 0.525 | $V_0 =$ 0.486 | $V_0 =$ 0.407 | $V_0 =$ 0.358 |
| Price in current (2019) United States Dollars (USD) | 7,644 | 63 | 63 | 63 | 63 | 63 | 63 |
| Upper limit of linear range (mg/L) | 1.599 | 0.000[a] | 1.399 | 1.000 | 1.399 | 1.000 | 1.800 |
| Upper limit of operational range (mg/L) | 1.399 | 1.599 | 1.399 | 1.399 | 1.399 | 1.399 | 0.201 |
| Calibration sensitivity (L/mg) using linear regression through the origin for the 0.000, 0.201, 0.400, and 0.601 mg/L standards | 0.127 | 0.107 | 0.085 | 0.102 | 0.087 | 0.079 | 0.051 |
| $R^2$ for the linear regression through the origin for the 0.000, 0.201, 0.400, and 0.601 mg/L standards | 0.99494 | 0.99876 | 0.99828 | 0.99904 | 0.99845 | 0.99914 | 0.98948 |
| Precision of standards (mg/L) | 0.01 | 0.005 | 0.006 | 0.009 | 0.01 | 0.01 | 0.03 |
| Estimated limit of detection (mg/L) | 0.01 | 0.006 | 0.008 | 0.01 | 0.01 | 0.02 | 0.03 |
| % Calibration check standard recovery | 96.8 | 100.9 | 100.0 | 101.2 | 101.5 | 104.0 | 103.6 |

[a]The 4 lowest concentrations of standard (0.000, 0.201, 0.400, and 0.601 mg/L) gave a quadratic effect coefficient that is different from 0 at the 95% confidence level (p-value = 0.0251); therefore, the calibration curve is nonlinear and the upper limit of linear range is 0.000 mg/L.

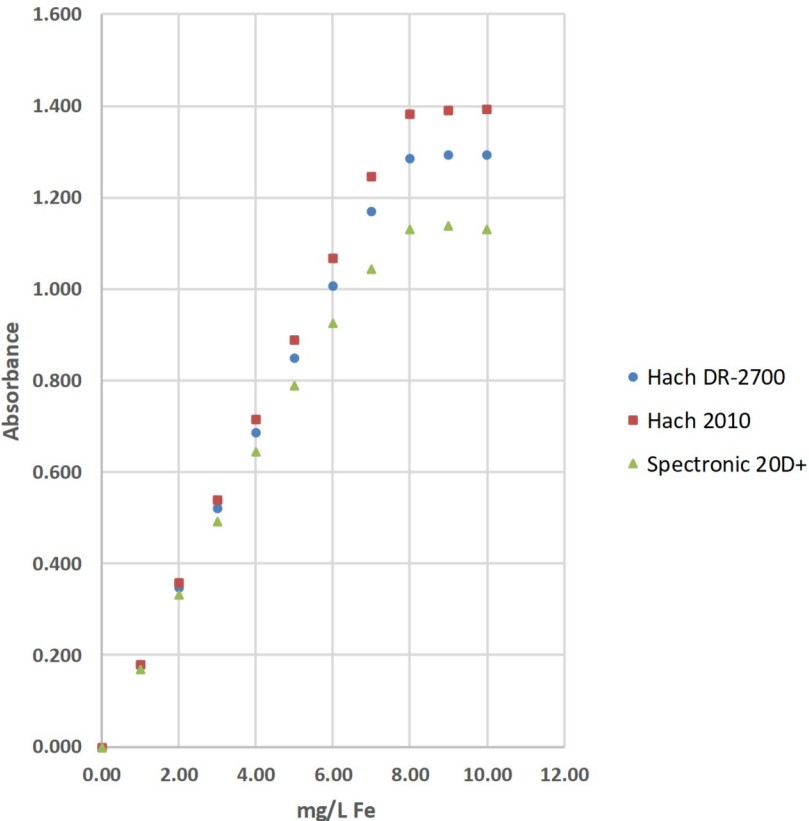

**Fig 8. The calibration results for the determination of total iron (Fe) by the 1,10-phenanthroline method using a Hach DR-2700, a Hach 2010, and a Thermal Electron Spectronic 20D+ spectrophotometer [36].**

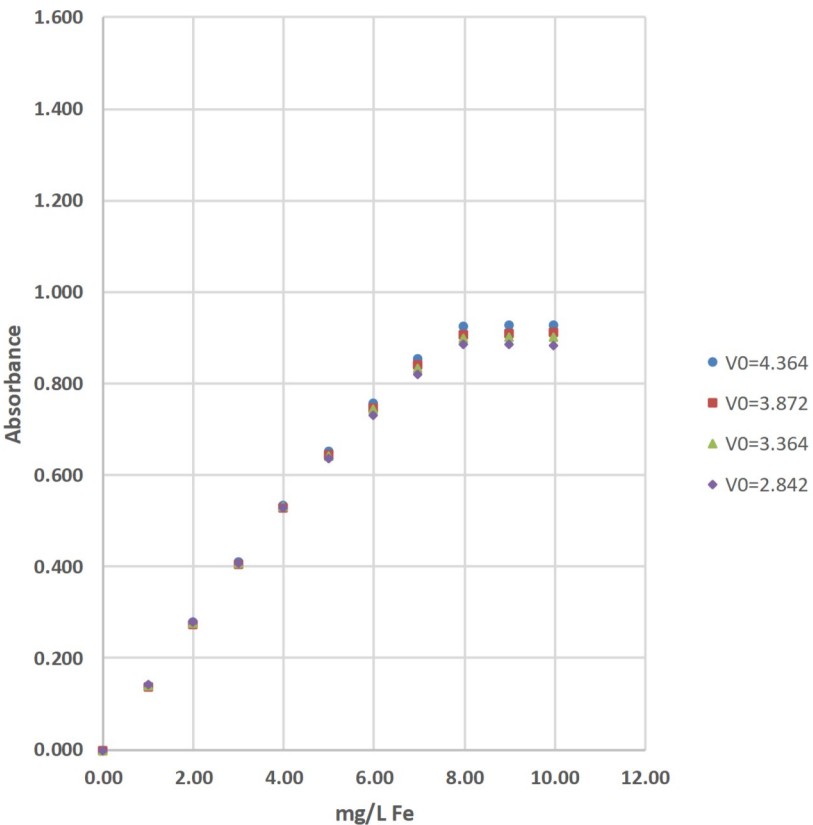

**Fig 9. The calibration results for the determination of total iron (Fe) by the 1,10-phenanthroline method using light emitting diode (LED) spectrophotometer #1 at blank voltages ($V_0$) = 4.364 V, 3.872 V, 3.364 V, and 2.842 V [36].**

## Total fluoride (F⁻) by the sodium 2-(parasulfophenylazo)-1,8-dihydroxy-3,6-naphthalene disulfonate method

The concentrations of total fluoride (F⁻) were determined by the sodium 2-(parasulfophenylazo)-1,8-dihydroxy-3,6-naphthalene disulfonate (SPADNS) method [36]. The absorbance of residual acid-zirconyl-SPADNS dye after the removal of colorless zirconium hexafluoride ($ZrF_6^{2-}$(aq, aqueous)) was measured at 567 nm with an Agilent Technologies Cary 60-UV-Vis (ultraviolet-visible) absorption spectrophotometer, and a LED spectrophotometer (Table 4; Figs 13 and 14). Standard solutions at 0.000, 0.201, 0.400, 0.601, 0.800, 1.000, 1.200, 1.399, 1.599, 1.800, and 2.000-mg/L of total F⁻(aq) were used to calculate linear range [25], operational range [36], and calibration sensitivity [6, 38]. Seven 0.000-mg/L standard solutions were analyzed as samples and used to calculate the precision of standards [36], and the estimated limit of detection [26]. Lastly, a 0.600-mg/L standard solution was analyzed as a sample and used to calculate the % calibration check standard recovery [36]. This work was done at the Indian Institute of Science (IISc) in Bangalore using the same LED spectrophotometer #1 that was used for the determinations of total Fe and total Mn at Norwich University in Northfield, VT.

### Adjusting blank voltage

The LED spectrophotometers use a variable resistor to adjust the voltage supplied to the LED (Table 1; Figs 3 and 5). This voltage ($V_0$, or the blank voltage) controls the brightness of the LED and was adjusted with a 0.00-mg/L Fe, 0.000-mg/L Mn, or 0.000-mg/L F⁻ standard in the

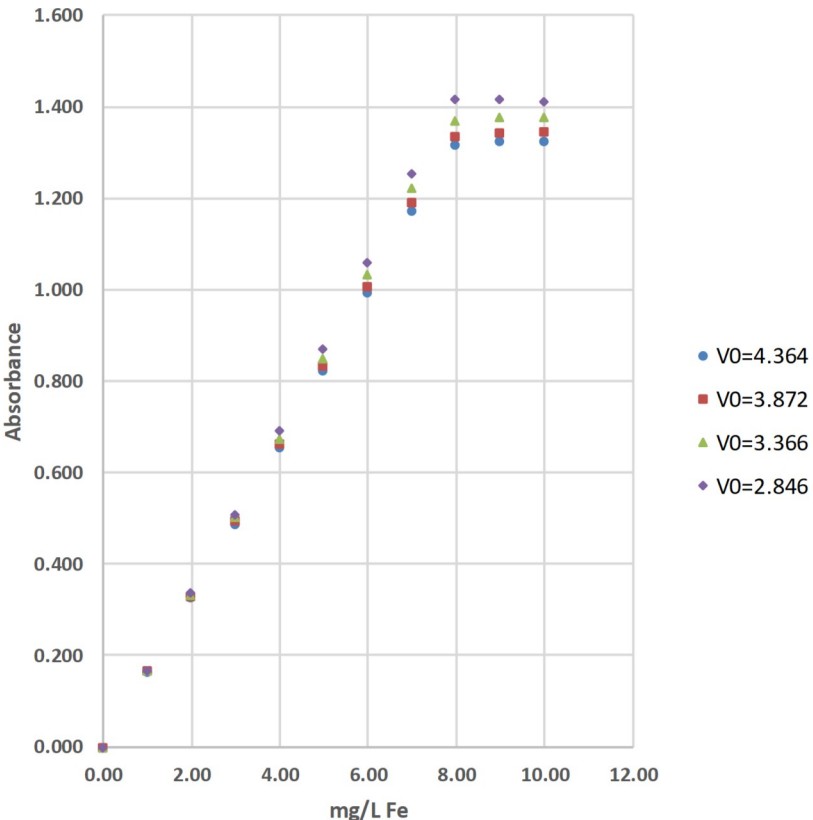

**Fig 10. The calibration results for the determination of total iron (Fe) by the 1,10-phenanthroline method using light emitting diode (LED) spectrophotometer #2 at blank voltages ($V_0$) = 4.364 V, 3.872 V, 3.366 V, and 2.846 V [36].**

sample cell holder (Figs 3 and 6). After it is adjusted, $V_0$ is held constant until all of the standards and samples in the analytical sequence are analyzed. For the total Fe and total Mn analyses in this study, 4 different values of $V_0$; that is, 4 different analytical sequences were used to evaluate the performance of a LED spectrophotometer and method (Tables 2 and 3; Figs 9, 10 and 12). More specifically, $V_0$ was initially set at or near its maximum voltage, then decreased in approximately 0.5-volt (V) increments until 4 different analytical sequences were performed (Tables 2 and 3; Figs 9, 10 and 12). For the total $F^-$ analyses in this study, 6 different values of $V_0$; that is, 6 different analytical sequences were used to evaluate the performance of a LED spectrophotometer and method (Table 4; Fig 14). More specifically, $V_0$ was initially set at or near its maximum voltage, then decreased in approximately 0.05-volt (V) increments until 6 different analytical sequences were performed (Table 4; Fig 14).

## Calculating absorbance

Two Fluke 175-True RMS Digital Multimeters were used to measure the voltages from the 2 LED spectrophotometers at Norwich University in Northfield, VT (Figs 3 and 7). A Haoyue DT830D Digital Multimeter was used to measure the voltages from the LED spectrophotometer at the Indian Institute of Science (IISc) in Bangalore. Eq 5 was used to convert these voltages to absorbances [39].

$$A \approx \log_{10} \frac{V_0}{V} \tag{5}$$

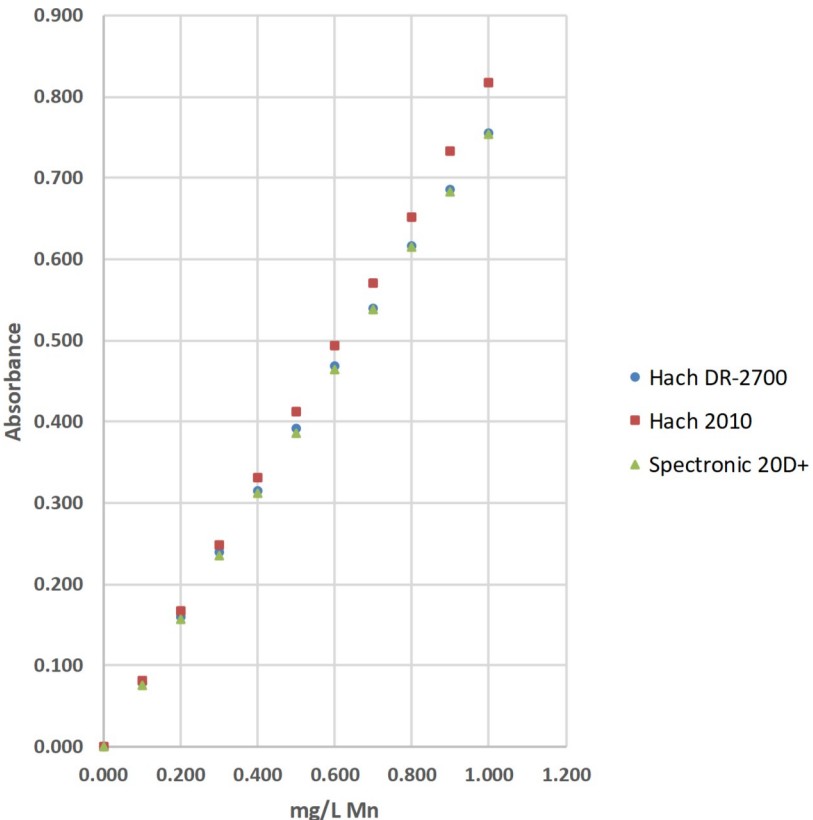

**Fig 11. The calibration results for the determination of total manganese (Mn) by the 1-(2-pyridylazo)-2-naphthol (PAN) method using a Hach DR-2700, a Hach 2010, and a Thermal Electron Spectronic 20D+ spectrophotometer** [37].

Where A is the absorbance, $V_0$ is the voltage with a 0.00-mg/L Fe, 0.000-mg/L Mn, or 0.000-mg/L F⁻ standard in the sample cell holder, and V is the voltage with a sample in the sample cell holder (Figs 3 and 6).

## Results and discussion

### Price

The price in current (2019) USD of the 4 commercial spectrophotometers used in this study ranged from $2,424 to $7,644 (Tables 2–4). In contrast, the price in current (2019) USD for a LED spectrophotometer is $63 for parts (Tables 2–4). The commercial spectrophotometers are significantly more expensive than the LED spectrophotometer (Tables 2–4).

 The $63 price of our LED spectrophotometer includes a single LED source assembly, and does not include the labor, overhead, and profit costs of the commercial spectrophotometers. Moreover, the LED spectrophotometer uses a simpler and less costly optical system. The LED spectrophotometer optical source costs approximately $3.27 for the source LED ($0.32), a 3.5-mm stereo male-male cable ($0.57), 1/8-inch heat shrink tubing ($0.10), and Sugru moldable glue ($2.28; Table 1). In contrast, a typical commercial spectrophotometer uses a more complicated and costly optical source. A typical commercial spectrophotometer might use a tungsten filament lamp, a field lens, a wavelength selector (an entrance slit, objective lens, diffraction grating, wavelength adjustment cam and actuator, light controller, occluder, and exit slit), a filter, and a measuring phototube [40]. As a result, our LED spectrophotometer uses a

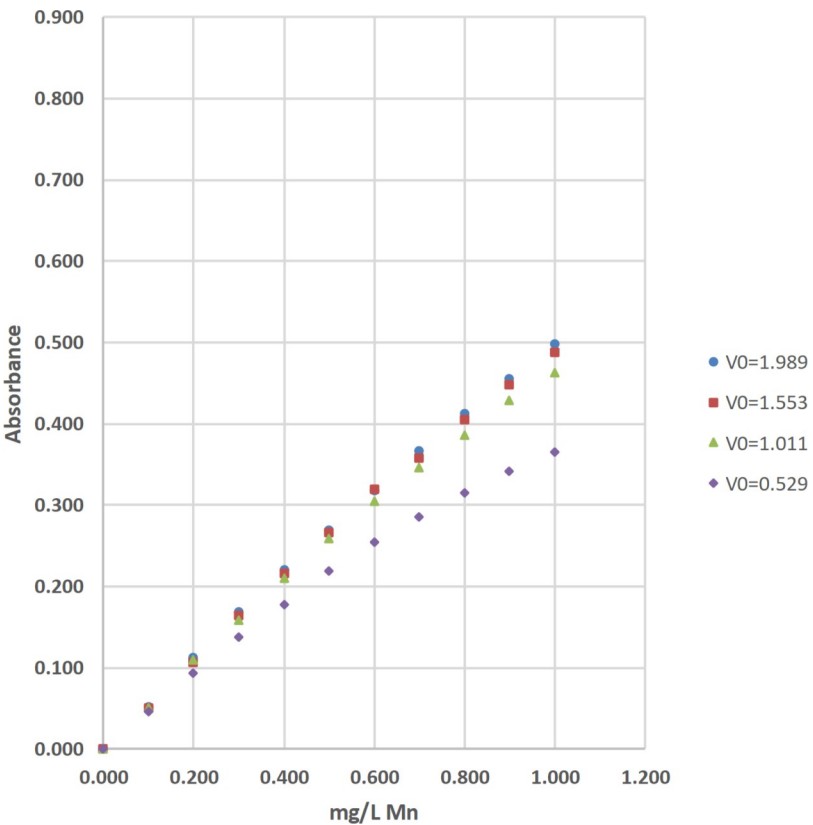

**Fig 12. The calibration results for the determination of total manganese (Mn) by the 1-(2-pyridylazo)-2-naphthol (PAN) method using light emitting diode (LED) spectrophotometer #1 at blank voltages ($V_0$) = 1.989 V, 1.553 V, 1.011 V, and 0.529 V [37].**

narrow range of polychromatic light with a spectral width of about 25 nm for a 510-nm source LED (Table 1). In contrast, commercial spectrophotometers use nearly monochromatic light [6]. Therefore, our spectrophotometer is less selective and more prone to interferences than a commercial spectrophotometer. Considering this limitation, it is imperative that users of our LED spectrophotometer be careful to avoid interferences by carefully monitoring the % recovery of known additions of standard to samples [1, 5, 36]. A known addition of standard to sample is also called a laboratory fortified sample matrix (LFSM), laboratory fortified matrix (LFM), or matrix spike (MS) [41]. Fortunately, this is standard operating procedure (SOP) in many routine drinking water testing laboratories [36, 41].

## Calibration

**Visual inspection and $R^2$ value of calibration curves.** Calibration curves are used in spectrophotometry to determine the concentrations of analytes in samples. In this study, the concentration of total Fe, total Mn, or total $F^-$ was the independent variable and was plotted on the x-axis. The absorbance was the dependent variable and was plotted on the y-axis (Figs 8–14). Calibration curves of the 1,10-phenanthroline method for total Fe and the 1-(2-pyridylazo)-2-naphthol (PAN) method for total Mn have positive slopes (Figs 8–12) [36, 37]. In contrast, calibration curves of the sodium 2-(parasulfophenylazo)-1,8-dihydroxy-3,6-naphthalene disulfonate (SPADNS) method for total $F^-$ have negative slopes (Figs 13 and 14) [36].

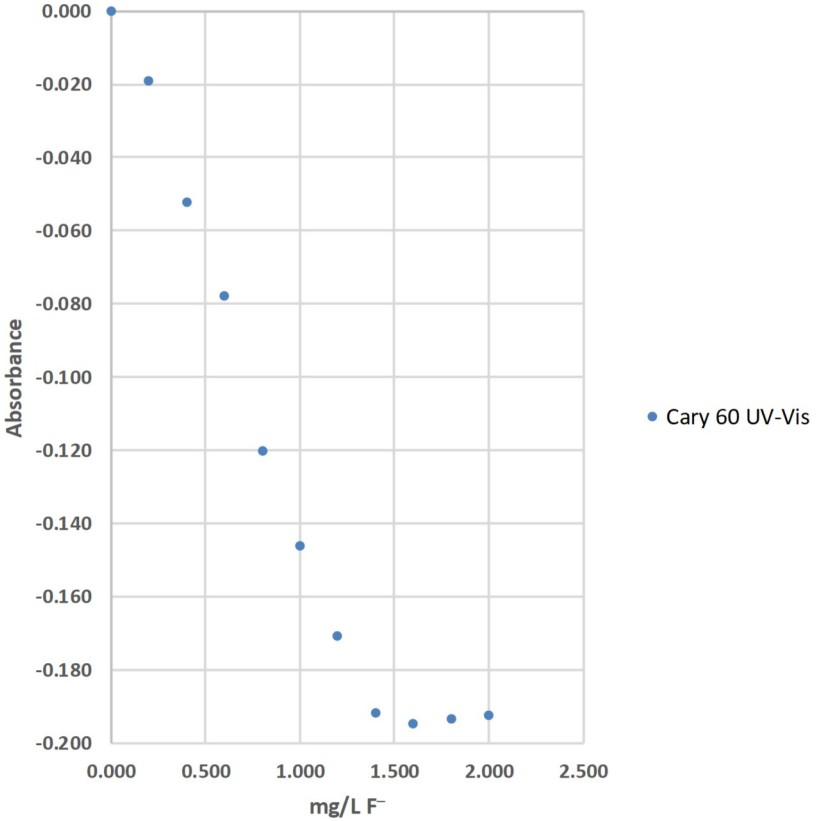

**Fig 13. The calibration results for the determination of total fluoride (F⁻) by the sodium 2-(parasulfophenylazo)-1,8-dihydroxy-3,6-naphthalene disulfonate (SPADNS) method using an Agilent Technologies Cary 60 UV-Vis (ultraviolet-visible) absorption spectrophotometer [36].**

According to the Beer-Bouguer-Lambert law, more commonly called Beer's law, at relatively low analyte concentrations, a linear calibration curve with the y-intercept going through the origin (0, 0) is expected [40]. For analyses with positive slopes and at relatively high analyte concentrations, little or no light from the source passes through the standard solutions and reach the detector; this causes a flattening of the calibration curve (Figs 8–10). For analyses with negative slopes and at relatively high analyte concentrations, an excess of light from the source passes through the standard solutions and reach the detector; this also causes a flattening of the calibration curve (Figs 13 and 14). Therefore, it is important to know the linearity and linear range for every method that is used on a given spectrophotometer. Historically, the linearity and the upper limit of linear range of calibration curves was subjectively evaluated by visual inspection (Figs 8–14).

A plot of $R^2$ values for each calibration curve in this study is shown in Fig 15. These $R^2$ values were used to subjectively evaluate the linearity of the 4 commercial and 2 LED spectrophotometers in this study (Fig 15). These calibration curves used linear regression through the origin for the 4 lowest concentration standards of each analyte (Tables 2–4).

The $R^2$ values ranged from 0.98948 to 1.00000 (Fig 15). These relatively large $R^2$ values suggest that all the calibration curves in this study are linear, especially considering that the calibration standard solutions have 3 significant figures. However, the importance of objectively testing the linearity of calibration curves is shown in the following example in which $R^2$ values are not associated with the linearity of the calibration curves (Table 4; Fig 15). The calibration

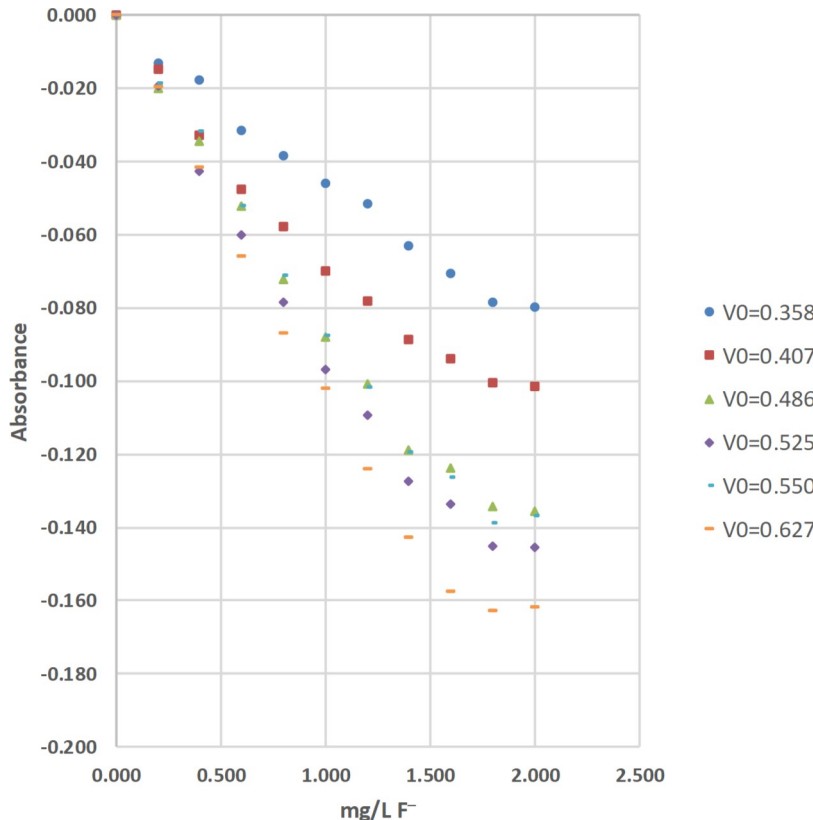

**Fig 14. The calibration results for the determination of total fluoride ($F^-$) by the sodium 2-(parasulfophenylazo)-1,8-dihydroxy-3,6-naphthalene disulfonate (SPADNS) method using light emitting diode (LED) spectrophotometer #1 at blank voltages ($V_0$) = 0.358 V, 0.407 V, 0.486 V, 0.525 V, 0.550 V, and 0.627 V [36].**

curve for LED spectrophotometer #1 at $V_0$ = 0.627 V had a statistically significant quadratic effect at the 95% confidence level (p-value = 0.0251); therefore, this calibration curve is non-linear despite having a relatively large $R^2$ (0.99876). In contrast, LED spectrophotometer #1 at $V_0$ = 0.358 V did not have a statistically significant quadratic effect at the 95% confidence level (p-value = 0.605); therefore, this calibration curve is linear despite having the smallest $R^2$ in the study (0.98948). In this example, the calibration curve with the larger $R^2$ is nonlinear and the calibration curve with the smaller $R^2$ is linear. This limitation of using $R^2$ as a subjective test of linearity is well known in analytical chemistry [42].

**Upper limit of linear range.** This paper demonstrates the novel use of objective tests for higher order polynomial relationships to measure the upper limits of linear range (Tables 2–4) [25]. These objective measurements of the upper limits of linear range greatly influenced the many design decisions that were made during the 3-year design and testing phase of this project.

Polynomial regression was used to objectively calculate the linear range of each calibration curve in this study (Tables 2, 3 and 4; Figs 8–14) [25]. Since Beer's law says that a linear calibration curve is expected, a regression of absorbance on the concentration of analyte and the concentration of analyte squared was used to test the significance of a second-order or quadratic effect [25]. If this quadratic effect was statistically significant at $\alpha$ = 0.05, the calibration curve is not linear at the 95% confidence level [25]. In contrast, if this quadratic effect was not statistically significant, the calibration curve is linear [25].

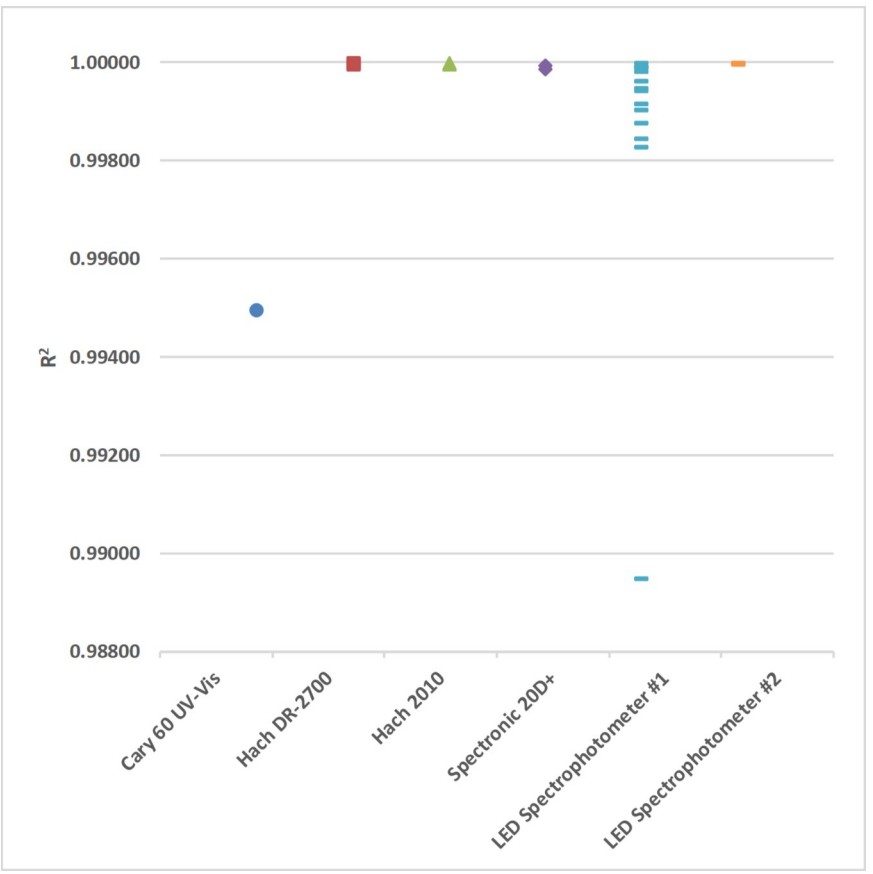

**Fig 15. The R$^2$ values for the Agilent Technologies Cary 60 UV-Vis (ultraviolet-visible), Hach DR-2700, a Hach 2010, a Thermal Electron Spectronic 20D+, and 2 light emitting diode (LED) spectrophotometers used in this study (Tables 2–4).**

At least 4 different concentrations of standard solution are required to test for a quadratic effect [43]. Therefore, the absorbances from the 4 lowest concentrations of standard solutions were tested for a significant quadratic effect [25]. If this quadratic effect was not statistically significant, the calibration curve is linear and the process was repeated with the absorbances from the 5 lowest concentrations of standard solutions [25]. This process continued until the quadratic effect was statistically significant [25].

For example, the regression of absorbance (0.000, 0.178, 0.349, and 0.521) on the respective concentration of total Fe (0.00, 1.01, 2.00, and 3.00 mg/L) and the respective concentration of total Fe squared (0.00, 1.02, 3.99, and 8.97 (mg/L)$^2$) was used to test the significance of a quadratic effect for the Hach DR-2700 (Table 2; Fig 8). The results of this first regression are shown in Eq 6.

$$\text{Absorbance} = -9.86 \times 10^{-4}(\text{mg/L})^2 + 0.177(\text{mg/L}) + 1.51 \times 10^{-4} \tag{6}$$

The quadratic effect coefficient ($-9.86 \times 10^{-4}$) is equivalent to 0 at the 95% confidence level (p-value = 0.213). It has a 95% confidence interval that includes 0; this interval extends from $-5.34 \times 10^{-3}$ to $3.37 \times 10^{-3}$. This conclusion suggests that the calibration curve is linear, so the process was repeated with the 0.00, 1.01, 2.00, 3.00, and 3.99-mg/L standard solutions. The absorbance equaled 0.686 at 3.99 mg/L and 15.94 (mg/L)$^2$. The results of this second regression

are shown in Eq 7.

$$\text{Absorbance} = -1.60 \times 10^{-3}(\text{mg/L})^2 + 0.178(\text{mg/L}) - 2.15 \times 10^{-4} \qquad (7)$$

In contrast, the quadratic effect coefficient ($-1.60 \times 10^{-3}$) is different from 0 at the 95% confidence level (p-value = 0.0343). It has a 95% confidence interval that does not include 0; this interval extends from $-2.91 \times 10^{-3}$ to $-2.91 \times 10^{-4}$. This conclusion suggests that the calibration curve is not linear, so the process was stopped and the upper limit of the linear range was set at 3.00 mg/L total Fe (Table 2; Fig 8).

The upper limits of linear range for the 3 commercial spectrophotometers used in the United States extended from 3.00 to 6.99 mg/L for total Fe (Table 2; Fig 8). By comparison, the upper limits of linear range for the 2 LED spectrophotometers extended from 3.00 to 4.99 mg/L for total Fe (Table 2; Figs 9 and 10). The upper limits of linear range for the 3 commercial spectrophotometers used in the United States extended from 0.500 to 0.700 mg/L for total Mn (Table 3; Fig 11). By comparison, the upper limits of linear range for the LED spectrophotometer extended from 0.400 to 0.600 mg/L for total Mn (Table 3; Fig 12). The upper limit of linear range for the 1 commercial spectrophotometer used in India was 1.599 mg/L for total $F^-$ (Table 4; Fig 13). By comparison, the upper limits of linear range for the LED spectrophotometer extended from 0.000 (nonlinear) to 1.800 mg/L for total $F^-$ (Table 4; Fig 14). The small blank voltages ($V_0$) of the total $F^-$ analyses most likely gave small signal to noise ratios and the large spread of upper limits of linear range for the LED spectrophotometer (Table 4; Fig 14). Even so, these results suggest that the LED spectrophotometers have upper limits of linear range that are generally comparable to those of the 4 commercial spectrophotometers used in this study (Tables 2–4; Figs 8–14) and would be sufficient for comparisons to World Health Organization (WHO) guidelines for drinking water quality or health-based values for these analytes [44, 45].

**Upper limit of operational range.** The operational or calibration range must be determined "before using a new method or instrument", according to the American Public Health Association (APHA), American Water Works Association (AWWA), and Water Environment Federation (WEF) [36]. These organizations say that both linear and nonlinear regions of a calibration curve can be used in routine drinking water testing laboratories [36]. Moreover, these organizations allow individual laboratories to make their own definition of operational range [36]. That is, there is no standard definition of operational range.

For this study, the upper limit of operational range was defined by the last change in the absorbance of incremental standards that is 50% less than the change in the absorbance of the corresponding 0.00- and 1.01-mg/L total Fe standards, the corresponding 0.00- and 0.100-mg/L total Mn standards, or the corresponding 0.000- and 0.201-mg/L of total $F^-$ standards (Tables 2–4; Figs 8–14). For example, the Hach DR-2700 gave absorbances of 0.000, 0.178, 1.169, 1.286, and 1.293 for the 0.00, 1.01, 6.99, 7.98, and 8.98-mg/L total Fe standards, respectively. The % change in absorbance for the 6.99- and 7.98-mg/L total Fe standards relative to the 0.00- and 1.01-mg/L total Fe standards is shown in Eq 8.

$$\frac{\text{Relative}}{\text{Change}} = \frac{(1.286 - 1.169)}{(0.178 - 0.000)} \times 100\% = 65.7\% \qquad (8)$$

In contrast, the % change in absorbance for the 7.98- and 8.98-mg/L total Fe standards relative to the 0.00- and 1.01-mg/L total Fe standards is shown in Eq 9.

$$\frac{\text{Relative}}{\text{Change}} = \frac{(1.293 - 1.286)}{(0.178 - 0.000)} \times 100\% = 3.93\% \qquad (9)$$

Therefore, the upper limit of the operational range was set at 7.98 mg/L total Fe (Table 2; Fig 8).

The upper limits of operational range for the 3 commercial spectrophotometers used in the United States and LED spectrophotometer #2 equaled 7.98 mg/L for total Fe (Table 2; Figs 8 and 9). By comparison, the upper limit of operational range for LED spectrophotometer #1 equaled 6.99 mg/L for total Fe (Table 2; Fig 9). The upper limits of operational range for all of the spectrophotometers was greater than or equal to ($\geq$) 1.000 mg/L for total Mn (Table 3; Figs 11 and 12). The upper limit of operational range for the 1 commercial spectrophotometer used in India was 1.399 mg/L for total $F^-$ (Table 4; Fig 13). By comparison, the upper limits of operational range for the LED spectrophotometer extended from 0.201 to 1.599 mg/L for total $F^-$ (Table 4; Fig 14). The small blank voltages ($V_0$) of the total $F^-$ analyses most likely gave small signal to noise ratios and the large spread of upper limits of operational range for the LED spectrophotometer (Table 4; Fig 14). Even so, these results suggest that the LED spectrophotometers have upper limits of operational range that are generally comparable to those of the 4 commercial spectrophotometers used in this study (Tables 2–4; Figs 8–14) and would be sufficient for comparisons to WHO guidelines for drinking water quality or health-based values for these analytes [44, 45].

**Calibration sensitivity and $R^2$.** Sensitivity is the ability to discriminate between small differences in analyte concentration [6]. More specifically, calibration sensitivity is the slope of a calibration curve [6]. That is, it is the change in absorbance per unit change in concentration. Therefore, the larger the slope, the larger the calibration sensitivity. The slope equals the absorptivity constant, a, which depends on the wavelength and the nature of the absorbing compound, multiplied by the path length through the absorbing compound, b [40]. This is Beer's law and is shown in Eq 10 [40].

$$A = abc = \log_{10} \frac{P_0}{P} \tag{10}$$

The absorbance, A, equals this slope (ab) multiplied by the concentration of absorbing compound, c [40]. The absorbance, A, also equals the log base 10 of the power of monochromatic light entering the sample, $P_0$, divided by the power of monochromatic light leaving the sample, P (Eq 10) [40]. Therefore, absorbance is a unitless ratio. In contrast, the units of slope or calibration sensitivity are L/mg.

For this study, calibration sensitivity was defined as the absolute value of the slope of a calibration curve using linear regression through the origin for the 0.00, 1.01, 2.00, and 3.00-mg/L standards for Fe (Table 2; Figs 8–10), for the 0.000, 0.100, 0.200, and 0.300-mg/L standards for Mn (Table 3; Figs 11 and 12), and for the 0.000, 0.201, 0.400, and 0.601-mg/L standards for $F^-$ (Table 4; Figs 13 and 14). These calibration sensitivities and the $R^2$ values for these linear regressions through the origin are shown in Tables 2–4.

The calibration sensitivities for the 3 commercial spectrophotometers used in the United States ranged from 0.165 to 0.180 L/mg for total Fe (Table 2; Fig 8). The $R^2$ values for these regressions ranged from 0.99994 to 1.00000 (Table 2; Fig 8). By comparison, the calibration sensitivities for the 2 LED spectrophotometers ranged from 0.136 to 0.169 L/mg for total Fe (Table 2; Figs 9 and 10). The $R^2$ values for these regressions ranged from 0.99980 to 0.99999 (Table 2; Figs 9 and 10).

The calibration sensitivities for the 3 commercial spectrophotometers used in the United States ranged from 0.781 to 0.828 L/mg for total Mn (Table 3; Fig 11). The $R^2$ values for these regressions ranged from 0.99987 to 0.99997 (Table 3; Fig 11). By comparison, the calibration sensitivities for the LED spectrophotometer ranged from 0.463 to 0.558 L/mg for total Mn

(Table 3; Fig 12). The $R^2$ values for these regressions ranged from 0.99943 to 0.99990 (Table 3; Fig 12).

The calibration sensitivity for the 1 commercial spectrophotometer used in India was 0.127 L/mg for total $F^-$ (Table 4; Fig 13). The $R^2$ for this regression was 0.99494 (Table 4; Fig 13). By comparison, the calibration sensitivities for the LED spectrophotometer ranged from 0.051 to 0.107 L/mg for total $F^-$ (Table 4; Fig 14). The $R^2$ values for these regressions ranged from 0.98948 to 0.99914 (Table 4; Fig 14).

These results suggest that the LED spectrophotometers have calibration sensitivities that are in general slightly less than, but still comparable to those of the 4 commercial spectrophotometers used in this study (Tables 2–4). The maximum $V_0$ of LED spectrophotometer #1 decreased significantly from approximately 4.364 V with the 510-nm LED (32 milliwatt/steradian; mW/sr) for total Fe, to 1.989 V with the 560-nm LED (0.022 mW/sr) for total Mn, and to 0.627 V with the 567-nm LED (0.73 mW/sr) for total $F^-$ (Tables 2–4). This decrease in $V_0$ is the most likely cause of this decrease in the calibration sensitivities of the LED spectrophotometers. If so, radiant intensity is an important criteria for selecting a source LED.

Moreover, the 2 LED spectrophotometers were made from the same components; however, the slight difference in the calibration sensitivities of these instruments for total Fe might result from slight differences in the performance of these components (Table 2; Figs 9 and 10).

## Precision

In analytical chemistry, precision is the agreement between repeated measurements of the same standard, known addition of standard to sample, or sample [36]. More specifically, the precision of standards is usually expressed as a sample standard deviation, *s* [46]. In this study, the precision of standards equals *s* for the 7 separately prepared 0.00-mg/L Fe standards that were analyzed as samples, the 7 separately prepared 0.000-mg/L Mn standards that were analyzed as samples, or the 7 separately prepared 0.000-mg/L $F^-$ standards that were analyzed as samples.

The precisions of standards for the 3 commercial spectrophotometers used in the United States ranged from 0.002 to 0.003 mg/L for total Fe (Table 2). By comparison, the precisions of standards for the 2 LED spectrophotometers ranged from 0.003 to 0.009 mg/L for total Fe (Table 2). The precisions of standards for the 3 commercial spectrophotometers used in the United States ranged from 0.0006 to 0.003 mg/L for total Mn (Table 3). By comparison, the precisions of standards for the LED spectrophotometer ranged from 0.001 to 0.005 mg/L for total Mn (Table 3). These results suggest that for total Fe and total Mn analyses, the LED spectrophotometers have precisions of standards that are in general slightly greater than, but still comparable to those of the 3 commercial spectrophotometers used in the United States (Tables 2 and 3). This conclusion is in agreement with that for the calibration sensitivities (Tables 2 and 3).

The precision of standards for the 1 commercial spectrophotometer used in India was 0.01 mg/L for total $F^-$ (Table 4). By comparison, the precisions of standards for the LED spectrophotometer ranged from 0.005 to 0.03 mg/L for total $F^-$ (Table 4). These results suggest that the LED spectrophotometer has precisions of standards that are generally comparable to those of the 1 commercial spectrophotometer used in India (Table 4).

## Limit of detection

The limit of detection is the smallest concentration that can be detected with reasonable certainty for a given analytical procedure [47]. There are many different ways to calculate limit of detection [48]. For this study, the estimated limit of detection was based on a 1-tailed 99%

confidence interval from the 7 separately prepared 0.00-mg/L Fe standards that were analyzed as samples, the 7 separately prepared 0.000-mg/L Mn standards that were analyzed as samples, or the 7 separately prepared 0.000-mg/L F⁻ standards that were analyzed as samples. This approach, called the method detection limit based on control charts, typically uses the results from many days, or even over all days that samples are analyzed [26]. Since this study used 1 day of analysis for each analyte, instead of the weeks, months, or even years of analyses used by the method detection limit based on control charts the sample mean, $\bar{X}$, for this study was set to 0 mg/L so that the resulting estimated limits of detection are comparable. This 1 day of analysis and setting $\bar{X}$ to 0 mg/L are why the resulting limits of detection are named estimated limits of detection in this study.

The estimated limits of detection for the 3 commercial spectrophotometers used in the United States ranged from 0.003 to 0.004 mg/L for total Fe (Table 2). By comparison, the estimated limits of detection for the 2 LED spectrophotometers ranged from 0.004 to 0.010 mg/L for total Fe (Table 2). The estimated limits of detection for the 3 commercial spectrophotometers used in the United States ranged from 0.0007 to 0.003 mg/L for total Mn (Table 3). By comparison, the estimated limits of detection for the LED spectrophotometer ranged from 0.001 to 0.006 mg/L for total Mn (Table 3). These results suggest that for total Fe and total Mn analyses, the LED spectrophotometers have estimated limits of detection that are in general slightly greater than, but still comparable to those of the 3 commercial spectrophotometers used in the United States (Tables 2 and 3). This conclusion is in agreement with that for the precisions of standards and the calibration sensitivities (Tables 2 and 3).

The estimated limit of detection for the 1 commercial spectrophotometer used in India was 0.01 mg/L for total F⁻ (Table 4). By comparison, the estimated limits of detection for the LED spectrophotometer ranged from 0.006 to 0.03 mg/L for total F⁻ (Table 4). These results suggest that the LED spectrophotometer has estimated limits of detection that are generally comparable to those of the 1 commercial spectrophotometer used in India (Table 4) and would be sufficient for comparisons to WHO guidelines for drinking water quality or health-based values for these analytes [44, 45].

## Instrument stability

A calibration check standard is a standard that is analyzed as a sample; typically, after every 10 samples and after the last sample are analyzed [36]. Calibration check standards are used to monitor the stability of the instrument and the method during the analysis of samples [36]. Ideally, the % calibration check standard recovery is 100% plus or minus an acceptable random error [36]. This error is often determined in routine drinking water testing laboratories by control charts; running 95% or 99% confidence intervals of the measured concentration or % recovery of the calibration check standards plotted on the y-axis, and the date of analysis plotted on the x-axis [36].

If a control chart of % calibration check standard recovery is not available, then a range of 100±15% calibration check standard recovery is acceptable [49]. All the % calibration check standard recoveries in this study are well within this 85-to-115% acceptable range (Tables 2–4).

The % calibration check standard recoveries for the 3 commercial spectrophotometers used in the United States ranged from 99.6 to 99.8% for total Fe (Table 2). By comparison, the % calibration check standard recoveries for the 2 LED spectrophotometers ranged from 98.8 to 100.0% for total Fe (Table 2). The % calibration check standard recoveries for the 3 commercial spectrophotometers used in the United States ranged from 98.8 to 107.7% for total Mn (Table 3). By comparison, the % calibration check standard recoveries for the LED

spectrophotometer ranged from 98.3 to 99.2% for total Mn (Table 3). The % calibration check standard recovery for the 1 commercial spectrophotometer used in India was 96.8% for total F⁻ (Table 4). By comparison, the % calibration check standard recovery for the LED spectrophotometer ranged from 100.0 to 104.0% for total F⁻ (Table 4). These results suggest that the LED spectrophotometers have % calibration check standard recoveries that are generally comparable to those of the 4 commercial spectrophotometers used in this study (Tables 2–4) and would be sufficient for comparisons to WHO guidelines for drinking water quality or health-based values for these analytes [44, 45].

### Bland-Altman comparison

Bland-Altman plots for pair-wise comparisons of LED spectrophotometer #1 with 3 commercial spectrophotometers (Hach DR-2700, Hach 2010, Thermal Electron Spectronic 20D+) for Fe and Mn, as well as for LED spectrophotometer #1 with Agilent Technologies Cary 60 UV-Vis for F⁻ are presented in Figs 16–22 [50].

LED spectrophotometer #1 was selected for these comparisons since it was used to measure every analyte (Tables 2–4). These comparisons were made with the LED spectrophotometer set at its maximum blank voltage ($V_0$; Tables 2–4). In each of these 7 comparisons, every point is within the lower and upper 95% confidence limits; therefore, the Bland-Altman plots show no statistically significant differences in the measurements made by the commercial spectrophotometers and LED spectrophotometer #1.

### Conclusions

Our LED spectrophotometer costs $63 USD for parts (Table 1). The 4 commercial spectrophotometers in this study ranged in cost from $2,424 to $7,644 USD (Tables 2–4). That is, our LED spectrophotometer costs between 2.2 and 0.70% of the 4 commercial spectrophotometers. Clearly, our LED spectrophotometer is extremely affordable compared to these 4 commercial spectrophotometers.

The upper limits of linear range for our LED spectrophotometer ranged from 3.00 to 4.99 mg/L, 0.400 to 0.600 mg/L, and 0.000 to 1.800 mg/L for the determinations of Fe, Mn, and F⁻,

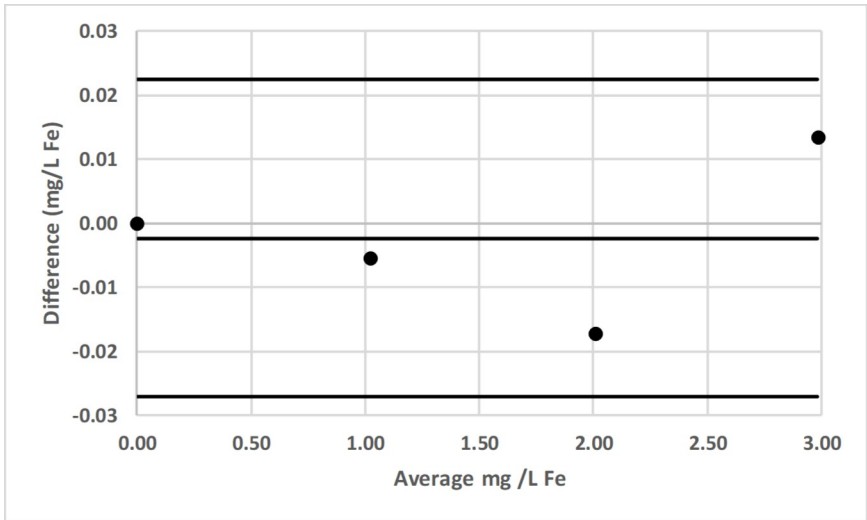

**Fig 16. Bland-Altman plot comparing the Hach DR-2700 and LED spectrophotometer #1 for the determination of total iron (Fe) by the 1,10-phenanthroline method [36].**

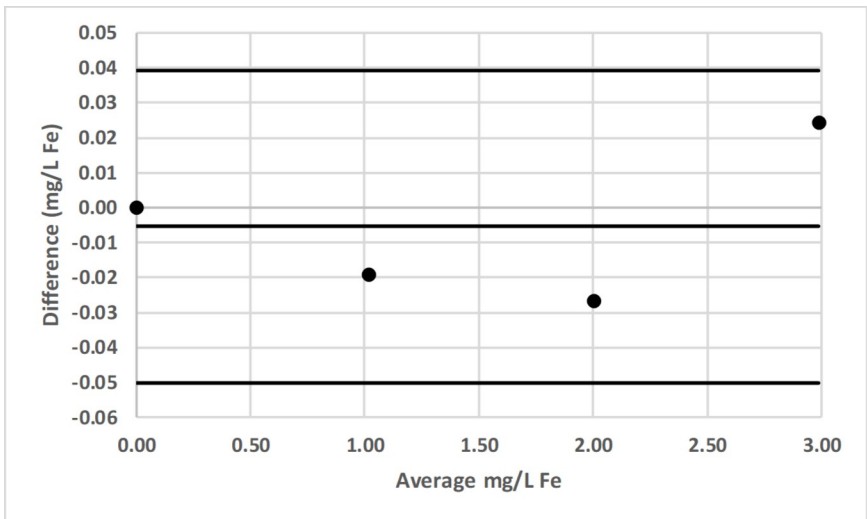

**Fig 17. Bland-Altman plot comparing the Hach 2010 and LED spectrophotometer #1 for the determination of total iron (Fe) by the 1,10-phenanthroline method** [36].

respectively (Tables 2–4). By comparison, these limits for the 4 commercial spectrophotometers in this study ranged from 3.00 to 6.99 mg/L, 0.500 to 0.700 mg/L, and 1.599 mg/L for the determinations of Fe, Mn, and F⁻, respectively (Tables 2–4). That is, each of these 3 ranges for our LED spectrophotometer overlapped with each of the 3 corresponding ranges for the commercial spectrophotometers. Therefore, there is no practical difference in the upper limits of linear range for our LED spectrophotometer and the 4 commercial spectrophotometers in this study; therefore, our LED spectrophotometer would be sufficient for comparisons to WHO guidelines for drinking water quality or health-based values for these analytes [44, 45].

Similarly, there are no practical differences in the upper limits of operational range, $R^2$ values, precisions of standards, and estimated limits of detection for our LED spectrophotometer and these 4 commercial spectrophotometers (Tables 2–4).

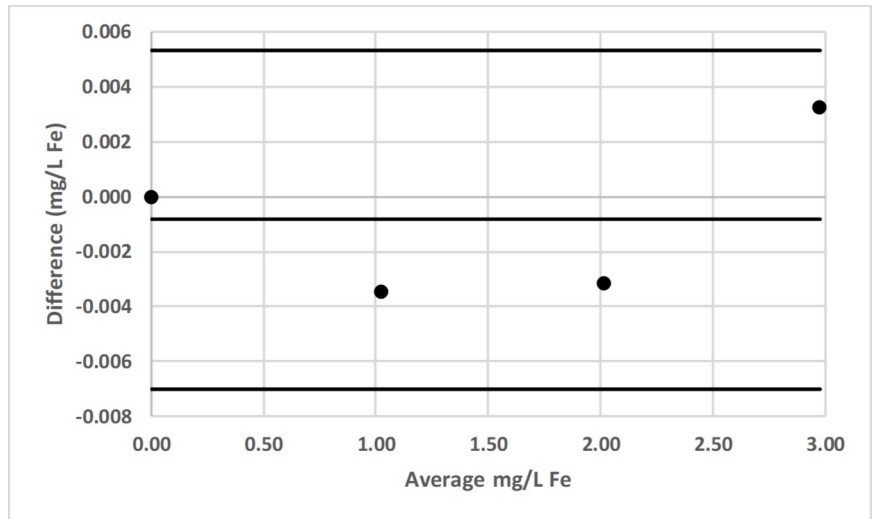

**Fig 18. Bland-Altman plot comparing the Thermal Electron Spectronic 20D+ and LED spectrophotometer #1 for the determination of total iron (Fe) by the 1,10-phenanthroline method** [36].

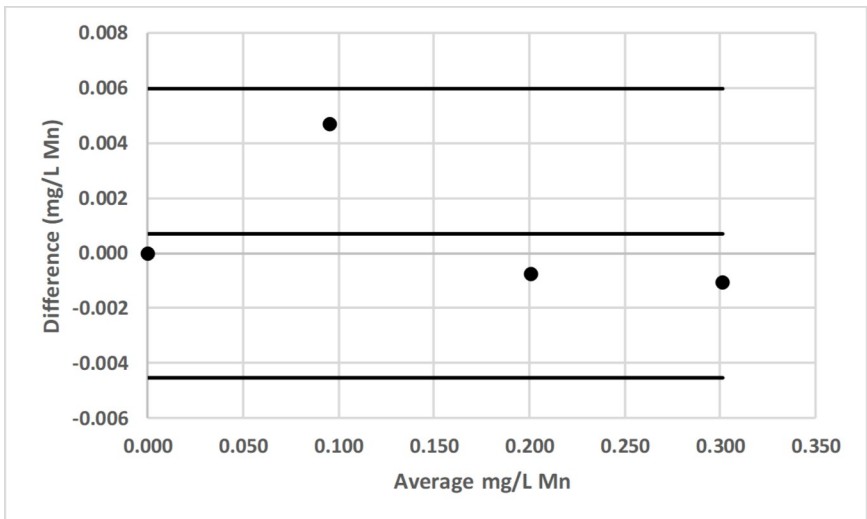

Fig 19. Bland-Altman plot comparing the Hach DR-2700 and LED spectrophotometer #1 for the determination of total manganese (Mn) by the Hach 1-(2-pyridylazo)-2-naphthol (PAN) method [37].

There is no practical difference in the calibration sensitivity for our LED spectrophotometer and these 4 commercial spectrophotometers for the determination of Fe; however, there is a practical difference in the calibration sensitivities for our LED spectrophotometer and these 4 commercial spectrophotometers for the determinations of Mn and F$^-$ (Tables 2–4). More specifically, the % of the average calibration sensitivity for our LED spectrophotometer divided by the average calibration sensitivity for these 4 commercial spectrophotometers are 87.7, 65.2, and 67.0% for the determinations of Fe, Mn, and F$^-$, respectively (Tables 2–4).

In conclusion, this paper describes the design, use, and performance of an extremely affordable, and sufficiently accurate and precise LED spectrophotometer for drinking water and other testing in regions with limited resources.

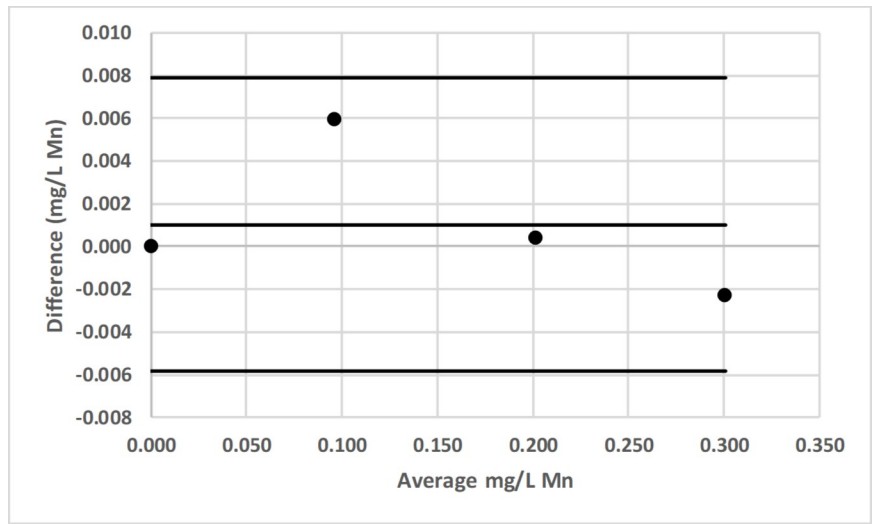

Fig 20. Bland-Altman plot comparing the Hach 2010 and LED spectrophotometer #1 for the determination of total manganese (Mn) by the Hach 1-(2-pyridylazo)-2-naphthol (PAN) method [37].

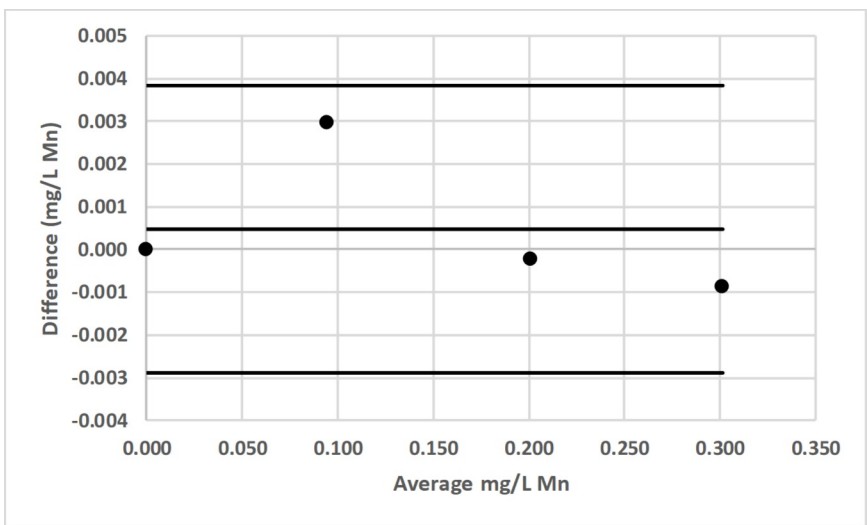

**Fig 21. Bland-Altman plot comparing the Thermal Electron Spectronic 20D+ and LED spectrophotometer #1 for the determination of total manganese (Mn) by the Hach 1-(2-pyridylazo)-2-naphthol (PAN) method [37].**

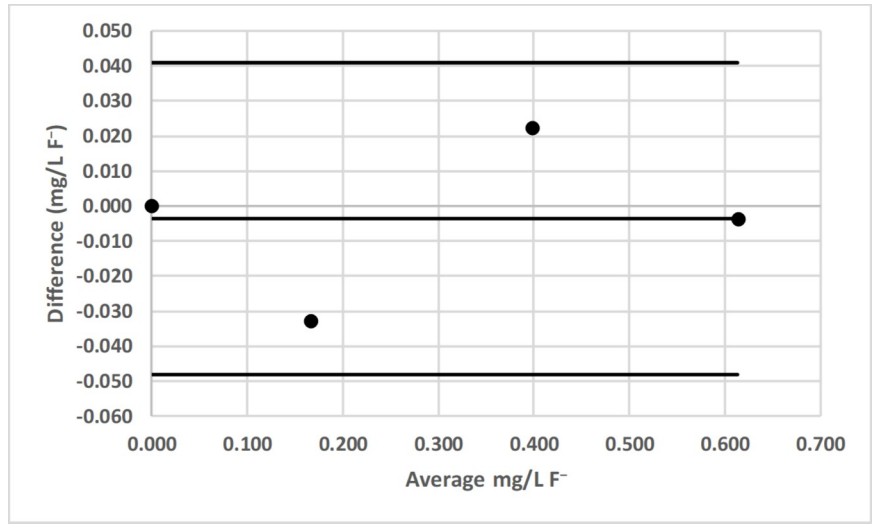

**Fig 22. Bland-Altman plot comparing the Agilent Technologies Cary 60 UV-Vis and LED spectrophotometer #1 for the determination of total fluoride (F⁻) by the sodium 2-(parasulfophenylazo)-1,8-dihydroxy-3,6-naphthalene disulfonate (SPADNS) method [36].**

## Supporting information

**S1 Tables. Iron data and analysis.**
(XLSX)

**S2 Tables. Manganese data and analysis.**
(XLSX)

**S3 Tables. Fluoride data and analysis.**
(XLSX)

**S4 Tables. The combined iron, manganese, and fluoride data and analysis.**
(XLSX)

## Acknowledgments

The authors thank Shubham Jain and Megha Rajasekhar for their valuable assistance with this project.

## Author Contributions

**Conceptualization:** Michael W. Prairie, Seth H. Frisbie, K. Kesava Rao.

**Formal analysis:** Seth H. Frisbie.

**Investigation:** Michael W. Prairie, Seth H. Frisbie, K. Kesava Rao, Anyamanee H. Saksri, Shreyas Parbat.

**Methodology:** Michael W. Prairie, Seth H. Frisbie, K. Kesava Rao.

**Project administration:** Michael W. Prairie, Seth H. Frisbie, K. Kesava Rao.

**Resources:** Michael W. Prairie, Seth H. Frisbie, K. Kesava Rao.

**Supervision:** Michael W. Prairie, Seth H. Frisbie, K. Kesava Rao.

**Validation:** Michael W. Prairie, Seth H. Frisbie, K. Kesava Rao.

**Visualization:** Michael W. Prairie, Seth H. Frisbie.

**Writing – original draft:** Seth H. Frisbie.

**Writing – review & editing:** Michael W. Prairie, Seth H. Frisbie, K. Kesava Rao, Anyamanee H. Saksri, Shreyas Parbat, Erika J. Mitchell.

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
