## [Decision Letter · Decision Letter 0]

16 Oct 2019

PONE-D-19-17629

An accurate, precise, and affordable light emitting diode spectrophotometer for drinking water and other testing with limited resources

PLOS ONE

Dear Frisbie,

Thank you for submitting your manuscript to PLOS ONE. After careful consideration, we feel that it has merit but does not fully meet PLOS ONE’s publication criteria as it currently stands. Therefore, we invite you to submit a revised version of the manuscript that addresses the points raised during the review process.

We would appreciate receiving your revised manuscript by Nov 30 2019 11:59PM. To enhance the reproducibility of your results, we recommend that if applicable you deposit your laboratory protocols in protocols.io, where a protocol can be assigned its own identifier (DOI) such that it can be cited independently in the future. For instructions see: http://journals.plos.org/plosone/s/submission-guidelines#loc-laboratory-protocols

We look forward to receiving your revised manuscript.

Kind regards,

Talib Al-Ameri, Ph.D

Academic Editor

PLOS ONE

Journal Requirements:

This study was supported by Norwich University, and the Indian Institute of Science.

The authors received no specific funding for this work.

Reviewers' comments:

Reviewer's Responses to Questions

**Comments to the Author**

1. Is the manuscript technically sound, and do the data support the conclusions?

Reviewer #1: Yes

Reviewer #2: No

2. Has the statistical analysis been performed appropriately and rigorously? 

Reviewer #1: Yes

Reviewer #2: No

3. Have the authors made all data underlying the findings in their manuscript fully available?

Reviewer #1: Yes

Reviewer #2: No

4. Is the manuscript presented in an intelligible fashion and written in standard English?

Reviewer #1: Yes

Reviewer #2: No

5. Review Comments to the Author

Reviewer #1: This manuscript describes the design and performance of an accurate and precise light emitting diode (LED) Spectrophotometer that costs approximately 63 U.S. dollars for parts, less than 2.6% of the cost of several commercial Spectrophotometers.

The paper is well organized and clearly written. The technical quality of the manuscript is satisfactory but the length is too long in relation to technical impact. Although reducing of equipment’s cost is not novel in most cases, this study considered as a valuable longitudinal study since the LED Spectrophotometer was continuously designed, tested, and improved over a three-year period. I would recommend publish this this manuscript after minor corrections.

Most regions of limited resources suffer from lack of availability of most components and that may affect the prices. I would recommend authors to consider this point in the motivation (in my knowledge these components are cheap in the industrial countries while the cost may be higher in regions of limited resources).

Secondly, I would suggest authors to involve statistical variability in the analysis (optional or in the future work).

Reviewer #2: In this paper, a light emitting diode spectrophotometer was designed and implemented for drinking water and other testing with limited resources. However, I recommend rejecting the paper for the following reasons.

1. The novelty of the paper is not well presented.

2. The contributions in the fields of study wholly missing.

3. The paper is not well structured and organized.

4. More unnecessarily information and details were presented in the paper.

5. The introduction section is very poor and did not highlight the problem statement in the field of study. In addition, the previous solutions in this field were not discussed or investigated in the study.

6. The related works section is missing.

7. The study presents well-known information, not new for specialists.

8. The “power unit and Light-emitting diode source circuit” they have been explained in detail though it is a simple electronic circuit.

9. Equation (11), this equation is well known for the standard deviation. Why the authors named it "precision standard".

10. Table 1 is not necessary to present the price of the components, instead, the authors can present the specifications of the proposed design.

11. Statistical analyses were not presented in a good manner.

12. The linear regression (R2) must be presented as a figure to allow the reader to seen the improvement in correlation between proposed LED spectrophotometer and commercial spectrophotometer.

13. The limits of agreement, better if it is measured under the Bland-Altman test. Where, the Bland-Altman test shows the upper and lower of agreement, difference of mean, standard deviation of difference. The difference here refers to the difference in measurement between proposed LED spectrophotometer and commercial spectrophotometer.

14. The resolution of Figs. 7-13.

15. The journal references are out of date.

16. Overall evolution, the paper is not well presented and organized, the contribution is missing, and the novelty is not clear. So, I recommend rejecting the paper for publication in a PLOS ONE Journal.

6. PLOS authors have the option to publish the peer review history of their article (what does this mean?). If published, this will include your full peer review and any attached files.

Reviewer #1: No

Reviewer #2: No

---

## [Author Response · Author response to Decision Letter 0]

29 Nov 2019

Dear Dr. Al-Ameri,

Thank you for reviewing our manuscript:

PONE-D-19-17629

An accurate, precise, and affordable light emitting diode spectrophotometer for drinking water and other testing with limited resources

We greatly appreciate the comments that you and the anonymous reviewers have made. These suggestions have proved very helpful in our revisions to the manuscript. We have addressed each of the reviewers’ comments, either through revisions to the manuscript, or provided further explanations in our rebuttal, as detailed below. 

As we revised this paper, we enlisted the assistance of an additional author, Dr. Erika J. Mitchell of Better Life Laboratories. Dr. Mitchell assisted us with updating the statistical analyses, the Bland-Altman plots, expanding the literature review, drafting portions of the revised introduction, and editing the paper. Accordingly, we would like to add her as an author to this manuscript.

Thank you again for your consideration of this manuscript,

Seth H. Frisbie, Ph.D.

Associate Professor of Chemistry

Department of Chemistry and Biochemistry

201 Juckett Hall

Norwich University

Northfield, VT 05663

Telephone: 1-802-485-2614

Fax: 1-802-485-2333

E-mail: sfrisbie@norwich.edu

PLOS ONE Decision: Revision required [PONE-D-19-17629] - [EMID:beb0e08c96cd9025]

em.pone.0.66a152.ac1b58a4@editorialmanager.com

on behalf of 

PLOS ONE <em@editorialmanager.com>

Wed 10/16/2019 4:59 AM

PONE-D-19-17629

An accurate, precise, and affordable light emitting diode spectrophotometer for drinking water and other testing with limited resources

PLOS ONE

Dear Frisbie,

Thank you for submitting your manuscript to PLOS ONE. After careful consideration, we feel that it has merit but does not fully meet PLOS ONE’s publication criteria as it currently stands. Therefore, we invite you to submit a revised version of the manuscript that addresses the points raised during the review process.

We would appreciate receiving your revised manuscript by Nov 30 2019 11:59PM. To enhance the reproducibility of your results, we recommend that if applicable you deposit your laboratory protocols in protocols.io, where a protocol can be assigned its own identifier (DOI) such that it can be cited independently in the future. For instructions see: http://journals.plos.org/plosone/s/submission-guidelines#loc-laboratory-protocols

• A rebuttal letter that responds to each point raised by the academic editor and reviewer(s). This letter should be uploaded as separate file and labeled 'Response to Reviewers'.

• A marked-up copy of your manuscript that highlights changes made to the original version. This file should be uploaded as separate file and labeled 'Revised Manuscript with Track Changes'.

• An unmarked version of your revised paper without tracked changes. This file should be uploaded as separate file and labeled 'Manuscript'.

We look forward to receiving your revised manuscript.

Kind regards,

Talib Al-Ameri, Ph.D

Academic Editor

PLOS ONE

Journal Requirements:

This study was supported by Norwich University, and the Indian Institute of Science.

Please remove any funding-related text from the manuscript and let us know how you would like to update your Funding Statement. 

REVISION MADE: We have removed the funding-related text from the Acknowledgments section. Please revise the Funding Statement for us since we cannot revise it from the document submission system.

Please update the Funding Statement to read:

The authors received no specific funding for this work. Non-specific support was provided by Norwich University and the Ministry of Human Resource Development, Government of India.

EJM’s affiliation is with Better Life Laboratories, a nonprofit organization that conducts scientific research and provides technical expertise, equipment, and training to help needy people around the world. Better Life Laboratories received no specific funding for this project from any donors. Donors to Better Life Laboratories provided no input in choosing the subject matter of this project, the materials selected to build the device, the brands or models of equipment selected for comparison, the method of analysis, the research findings, or the manner of disseminating the results. This does not alter our adherence to PLOS ONE policies on sharing data and materials.

Reviewers' comments:

Reviewer's Responses to Questions

Comments to the Author

1. Is the manuscript technically sound, and do the data support the conclusions?

Reviewer #1: Yes

Reviewer #2: No

2. Has the statistical analysis been performed appropriately and rigorously? 

Reviewer #1: Yes

Reviewer #2: No

3. Have the authors made all data underlying the findings in their manuscript fully available?

Reviewer #1: Yes

Reviewer #2: No

4. Is the manuscript presented in an intelligible fashion and written in standard English?

Reviewer #1: Yes

Reviewer #2: No

5. Review Comments to the Author

Reviewer #1: This manuscript describes the design and performance of an accurate and precise light emitting diode (LED) Spectrophotometer that costs approximately 63 U.S. dollars for parts, less than 2.6% of the cost of several commercial Spectrophotometers.

The paper is well organized and clearly written. The technical quality of the manuscript is satisfactory but the length is too long in relation to technical impact. Although reducing of equipment’s cost is not novel in most cases, this study considered as a valuable longitudinal study since the LED Spectrophotometer was continuously designed, tested, and improved over a three-year period. I would recommend publish this this manuscript after minor corrections.

Most regions of limited resources suffer from lack of availability of most components and that may affect the prices. I would recommend authors to consider this point in the motivation (in my knowledge these components are cheap in the industrial countries while the cost may be higher in regions of limited resources).

REVISION MADE: We have added additional text to the Materials and Methods section to explain this point.

Secondly, I would suggest authors to involve statistical variability in the analysis (optional or in the future work).

REVISION MADE: We have expanded our section explaining our statistical analysis to make the analysis more clear.

Reviewer #2: In this paper, a light emitting diode spectrophotometer was designed and implemented for drinking water and other testing with limited resources. However, I recommend rejecting the paper for the following reasons.

1. The novelty of the paper is not well presented.

REVISION MADE: We have expanded the Introduction, and the Results and Discussion sections to more thoroughly explain the novelty of the paper. 

2. The contributions in the fields of study wholly missing.

REVISION MADE: We have expanded the Introduction, and the Results and Discussion sections to address this comment. We now include a more thorough literature review of the design, construction, and testing of other LED spectrophotometers in education and industry. These new expanded sections include 23 new references.

3. The paper is not well structured and organized.

REVISION MADE: We have expanded the introduction, literature review, and Results and Discussion sections. We have also made the section headings more descriptive. We believe this improved the structure and organization of the paper.

4. More unnecessarily information and details were presented in the paper.

REVISION MADE: We have revised the paper to make it more clear why we have chosen to present these details.

5. The introduction section is very poor and did not highlight the problem statement in the field of study. In addition, the previous solutions in this field were not discussed or investigated in the study.

REVISION MADE: We have expanded the Introduction section to highlight the goals of the study as well as previous related works as presented in the literature. We have also added an explanation about how our study is novel.

6. The related works section is missing.

REVISION MADE: We have expanded the Introduction to discuss previous related works. This new expanded Introduction includes 22 new references.

7. The study presents well-known information, not new for specialists.

A major goal and a novel aspect of our paper is that we give enough information so that our spectrophotometer can be built, tested, and used by non-specialists. Our goal is to make all aspects of the design comprehensible to laboratory analytical chemists as well as electrical engineers. Thus, we do not assume any prior background knowledge about components of the design, even though the details may be familiar to some readers such as electrical engineers.

8. The “power unit and Light-emitting diode source circuit” they have been explained in detail though it is a simple electronic circuit.

A major goal and a novel aspect of our paper is to provide enough information so that our spectrophotometer can be built, tested, and used by non-specialists. For our paper to be effective, we must provide sufficient detail so that a non-specialist in a resource-limited country or area has enough information to build, test, and use our LED spectrophotometer.

9. Equation (11), this equation is well known for the standard deviation. Why the authors named it "precision standard".

REVISION MADE: Equation 11 and its associated text have been deleted.

There are 3 different measures of precision in analytical chemistry: the precision of standards, the precision of samples, and the precision of known additions of standard to samples. The precision of standards is defined as the sample standard deviation from the repeated analyses of a single standard solution.

10. Table 1 is not necessary to present the price of the components, instead, the authors can present the specifications of the proposed design.

REVISION MADE: We have included a new Figure (new Figure 1) to present the specifications of the design. However, we have also retained Table 1 as we believe it provides essential information to readers about which specific components that we used. It also provides a clear explanation of how we calculated the cost of $63 USD that we mention in the text.

11. Statistical analyses were not presented in a good manner.

REVISION MADE: R2 and Bland-Altman plots have been added as requested. We have also expanded our explanation of using tests for higher order polynomial relationships to objectively test the linearity of calibration curves and to measure the upper limits of linear range in this study, and using control charts to estimate detection limits.

12. The linear regression (R2) must be presented as a figure to allow the reader to seen the improvement in correlation between proposed LED spectrophotometer and commercial spectrophotometer.

REVISION MADE: As requested, the R2 values for every spectrophotometer and every analyte are shown together in a new figure, Fig 15.

13. The limits of agreement, better if it is measured under the Bland-Altman test. Where, the Bland-Altman test shows the upper and lower of agreement, difference of mean, standard deviation of difference. The difference here refers to the difference in measurement between proposed LED spectrophotometer and commercial spectrophotometer.

REVISION MADE: Bland-Altman plots for each pairwise comparison of LED spectrophotometer #1 and the commercial spectrophotometers for each analyte have been added.

14. The resolution of Figs. 7-13.

As specified by the journal, these figures are saved as TIFF with 300 dpi resolution. However, the figures may have been blurred when they were incorporated in the PDF for review. Clear 300 dpi images for all figures have been uploaded to the journal, but we cannot control their appearance in the preview PDF files.

15. The journal references are out of date.

REVISION MADE: We have expanded our literature review and added 23 new references, many of which are very recent (only 1 or 2 years old).

16. Overall evolution, the paper is not well presented and organized, the contribution is missing, and the novelty is not clear. So, I recommend rejecting the paper for publication in a PLOS ONE Journal.

We are very grateful to Reviewer #2 for thoughtfully reading our article and us giving very helpful feedback. Following the specific comments, we have improved the presentation, organization, and contribution to the field and more clearly explained the novel aspects of this project. The authors all agree that the current version of our manuscript is greatly improved as a result. Thank you very much for sharing your time and expertise.

6. PLOS authors have the option to publish the peer review history of their article (what does this mean?). If published, this will include your full peer review and any attached files.

Do you want your identity to be public for this peer review? For information about this choice, including consent withdrawal, please see our Privacy Policy.

Reviewer #1: No

Reviewer #2: No

---

## [Decision Letter · Decision Letter 1]

6 Dec 2019

An accurate, precise, and affordable light emitting diode spectrophotometer for drinking water and other testing with limited resources

PONE-D-19-17629R1

Dear Dr. Frisbie,

We are pleased to inform you that your manuscript has been judged scientifically suitable for publication and will be formally accepted for publication once it complies with all outstanding technical requirements.

With kind regards,

Talib Al-Ameri, Ph.D

Academic Editor

PLOS ONE

Additional Editor Comments (optional):

Reviewers' comments:

Reviewer's Responses to Questions

**Comments to the Author**

1. If the authors have adequately addressed your comments raised in a previous round of review and you feel that this manuscript is now acceptable for publication, you may indicate that here to bypass the “Comments to the Author” section, enter your conflict of interest statement in the “Confidential to Editor” section, and submit your "Accept" recommendation.

Reviewer #2: All comments have been addressed

2. Is the manuscript technically sound, and do the data support the conclusions?

Reviewer #2: Yes

3. Has the statistical analysis been performed appropriately and rigorously? 

Reviewer #2: Yes

4. Have the authors made all data underlying the findings in their manuscript fully available?

Reviewer #2: Yes

5. Is the manuscript presented in an intelligible fashion and written in standard English?

Reviewer #2: Yes

6. Review Comments to the Author

Reviewer #2: I have carefully read the corrections that the authors did it. All my comments have been addressed, therefore I recommend to accept the paper without more corrections.

7. PLOS authors have the option to publish the peer review history of their article (what does this mean?). If published, this will include your full peer review and any attached files.

Reviewer #2: Yes: Sadik Kamel Gharghan

---

## [Editor Report · Acceptance letter]

6 Jan 2020

PONE-D-19-17629R1 

An accurate, precise, and affordable light emitting diode spectrophotometer for drinking water and other testing with limited resources 

Dear Dr. Frisbie:

I am pleased to inform you that your manuscript has been deemed suitable for publication in PLOS ONE. Congratulations! Your manuscript is now with our production department. 

With kind regards,

on behalf of

Dr. Talib Al-Ameri 

Academic Editor

PLOS ONE